# Ticks parasitised feathered dinosaurs as revealed by Cretaceous amber assemblages

Enrique Peñalver [1], Antonio Arillo[2], Xavier Delclòs [3], David Peris [4], David A. Grimaldi[5], Scott R. Anderson [6], Paul C. Nascimbene[5] & Ricardo Pérez-de la Fuente[7]

Ticks are currently among the most prevalent blood-feeding ectoparasites, but their feeding habits and hosts in deep time have long remained speculative. Here, we report direct and indirect evidence in 99 million-year-old Cretaceous amber showing that hard ticks and ticks of the extinct new family Deinocrotonidae fed on blood from feathered dinosaurs, non-avialan or avialan excluding crown-group birds. A †*Cornupalpatum burmanicum* hard tick is entangled in a pennaceous feather. Two deinocrotonids described as †*Deinocroton draculi* gen. et sp. nov. have specialised setae from dermestid beetle larvae (hastisetae) attached to their bodies, likely indicating cohabitation in a feathered dinosaur nest. A third conspecific specimen is blood-engorged, its anatomical features suggesting that deinocrotonids fed rapidly to engorgement and had multiple gonotrophic cycles. These findings provide insight into early tick evolution and ecology, and shed light on poorly known arthropod–vertebrate interactions and potential disease transmission during the Mesozoic.

[1] Museo Geominero, Instituto Geológico y Minero de España, 28003 Madrid, Spain. [2] Departamento de Zoología y Antropología Física, Facultad de Biología, Universidad Complutense, 28040 Madrid, Spain. [3] Departament de Dinàmica de la Terra i de l'Oceà and Institut de Recerca de la Biodiversitat (IRBio), Facultat de Ciències de la Terra, Universitat de Barcelona, 08028 Barcelona, Spain. [4] Departament de Ciències Agràries i del Medi Natural, Universitat Jaume I, 12071 Castelló de la Plana, Spain. [5] Division of Invertebrate Zoology, American Museum of Natural History, New York, NY 10021, USA. [6] Independent Researcher, Moon Township, USA. [7] Oxford University Museum of Natural History, Parks Road, Oxford OX1 3PW, UK. Correspondence and requests for materials should be addressed to E.P. (email: e.penalver@igme.es) or to R.P.-d.l.F. (email: ricardo.perez-de-lafuente@oum.ox.ac.uk)

Fossils of ectoparasitic, haematophagous arthropods associated with integumentary remains of their vertebrate hosts are scarce and were restricted to the Cainozoic: feather remains in the gut content of an Eocene bird louse[1], lice eggs attached to several hairs in Eocene amber[2], a hard tick (Ixodidae) adjacent to a coprolite and a hair in Miocene amber[3], and a flea preserved together with several mammalian hairs in Miocene amber[4]. Likewise, Mesozoic ticks have a poor fossil record that has hindered understanding of the early evolution of these blood-sucking ectoparasites. Modern ticks are classified into three families: Nuttalliellidae, Argasidae (soft ticks), and Ixodidae. Nuttalliellidae, known from a single, extant species, *Nuttalliella namaqua*, is considered the closest extant relative to the ancestral tick lineage, bearing a mix of autapomorphies (e.g., ball and socket leg joints) and plesiomorphies, and appears to be the sister group to the clade (Ixodidae + Argasidae) based on morphological and molecular studies[5, 6].

Here, we present the fossil record of an ectoparasitic individual in intimate association with integumentary remains of its host—a hard tick entangled in a pennaceous feather preserved in ca. 99 million-year-old Burmese amber[7]. Additionally, tick specimens of a new family, also found in Burmese amber, may be indirectly related to feathered dinosaur hosts due to the presence of specialised setae from dermestid beetle larvae (hastisetae) attached to the ticks, along with further evidence of taphonomic nature, both indicating resin entrapment in close proximity to the host's nest.

## Results

<div align="center">

Arachnida Lamarck, 1801[8]
Parasitiformes Reuter, 1909[9]
Ixodida Leach, 1815[10]
Ixodidae Dugès, 1834[11]
*Cornupalpatum burmanicum* Poinar and Brown, 2003[12]

</div>

**Remarks**. The specimen AMNH Bu JZC-F18, preserved in Burmese amber, is a nymph based on its eight legs and absent genital pore (Figs. 1, 2). The tick, ca. 0.9 mm long from the posterior margin to the apex of hypostome, has ventrolateral claws on palpomere III, lacks eyes, has all coxae with spurs, and shows 11 festoons (Figs. 1, 2; Supplementary Fig. 2a). Within the current diversity of Cretaceous hard ticks, none of them described as a nymph, these characters classify AMNH Bu JZC-F18 within *Cornupalpatum burmanicum*, described on the basis of two larvae[12]. The scutum, the teeth in the hypostome, the Haller's organ, and the striate integument were not visible in the holotype of *C. burmanicum*, likely due to the specimen's state of preservation. In addition, the new specimen does not fit some of the characters in the original description of the species, some of which could represent ontogenetic variation: the ventrolateral claws in the third palpal segment are less developed, the central festoon is as wide as the others (not narrower), and the second palpal segment is more elongated. In any case, we acknowledge that *C. burmanicum* and *Compluriscutula vetulum*, the other Cretaceous ixodid species based on a larval stage[13], show a high degree of similarity with ticks of the extant genus *Amblyomma*[14], and a Cretaceous species within that genus based on an adult was recently named[15]. A future revision of the described Cretaceous hard ticks re-evaluating all the critical characters is necessary to elucidate their relationships.

Most significantly, the hard tick has one leg entangled in the barb of a pennaceous feather with a rather thick rachis basally (Fig. 1; Supplementary Fig. 1). Its preserved section is 19.4 mm long and shows over 50 preserved barbs, most of them attached to

the rachis, but with their apices lost at the surface of the amber. Those barbs that happen to be complete are much shorter on one side of the preserved rachis section than those on the other side (ca. 11 vs. 19.5 mm). Some barbs show damage, which likely occurred before having become embedded in the resin (Supplementary Fig. 1a). The fine preservation of the barbules allows us to distinguish their blade-like bases and their pennula, which display spined nodes and internodes. Most nodes in a distal position along the barbs are well defined and show short spines that are (sub)equally developed on both sides of the barbule pennulum (Fig. 1d; Supplementary Fig. 1c, d). Some poorly defined nodes present in more proximal–medial areas of the barb, however, show relatively long spines on one side of the pennulum that form hooklets (=hamuli) (Supplementary Fig. 1e, f). In addition, two isolated barbs from a different feather are close to the semicomplete one (Supplementary Fig. 1b), and a detached pennulum showing hooklets on one of its sides, ca. 0.6 mm long, is also present in the amber piece (Fig. 1f). Pigments indicating colour patterns have not been observed.

<div align="center">

Deinocrotonidae Peñalver, Arillo, Anderson and Pérez-de la Fuente fam. nov.

</div>

**Type genus**. *Deinocroton* gen. nov. Monotypic.
**Etymology**. From Greek *deinos*, "terrible", and *krotốn*, "tick". Gender: neutral.
**Diagnosis (both sexes)**. Integument with closely spaced, deep pits, and mound-like elevations between pits; integument not convoluted, lacking microsculpture. Pseudoscutum distinct (abbreviated in females), pitted but without elevations. Eyes absent. Hypostome subterminal. Basis capituli not bordered by coxae I. Palpi elongated, gracile; palpomere II distally thickened and bent in ventral direction, palpomeres III and IV elongated, tubular, fully mobile. Genital aperture transverse, close to the capitulum in males and slightly posteriad in females. Presence of a conspicuous anteroventral depressed area, post-genital in position. Spiracles smooth, medium sized, located at the level of coxae IV. Genital groove distinct, medially divided in two sections and extending posteriorly. Anal pore terminal. Preanal groove prolonged posteriorly, with sides closing. Legs ruffled. All coxae with short spurs in rows. Leg joints not of the ball and socket type but notch-like processes present. Haller's organ proximal capsule completely open. Festoons absent.

*Deinocroton draculi* Peñalver, Arillo, Anderson and Pérez-de la Fuente gen. et sp. nov.

*Etymology*. Patronym for the main character of the gothic horror novel by Irish writer Abraham "Bram" Stoker, which is a fictionalised account of Vlad III, or Vlad Dracula (ca. 1429–1476).

**Holotype**. Adult male (AMNH Bu-SA5a), ca. 3.9 mm long from posterior margin to apex of hypostome (Figs. 3a, e–g, j, k, 4a, f–h, 5a, c, d, f, g; Supplementary Fig. 2c, e).
**Additional material**. Allotype: female (CM 63007) (Fig. 4b, c; Supplementary Figs. 2d, 3). Paratypes: male (AMNH Bu-SA5b) (Figs. 3a, d, i, l, 4d, e, 5b, e; Supplementary Fig. 2b) and engorged female (CM 63001) (Figs. 3b, c, h, m, 5h, i). All adults (see Supplementary Note 1 for more details).
**Locality and horizon**. Southwest of Tanai (close to Maingkhwan village) in the Hukawng Basin, Kachin State area (northern Myanmar), likely from the Noije bum opencast system of mines; earliest Cenomanian[7].
**Diagnosis for genus and species**. As for the family.

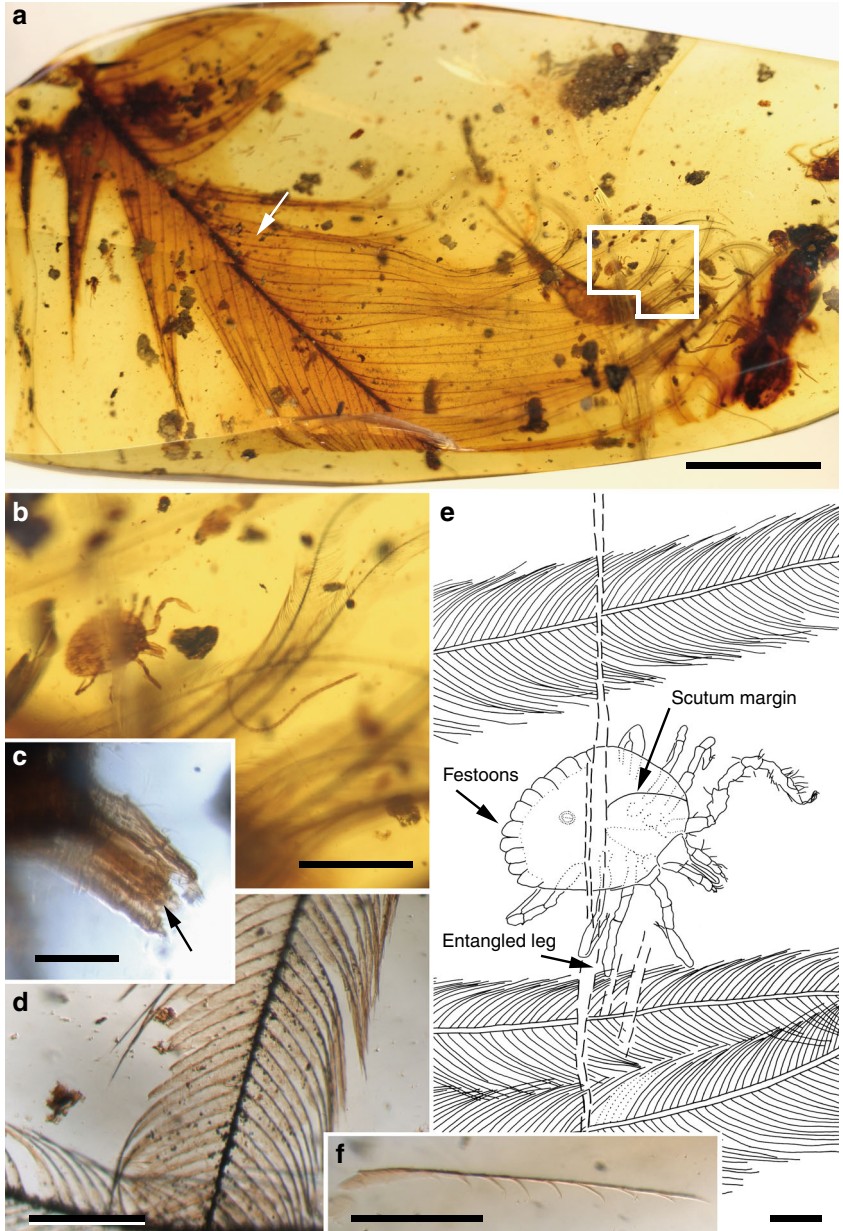

**Fig. 1** *Cornupalpatum burmanicum* hard tick entangled in a feather. **a** Photograph of the Burmese amber piece (Bu JZC-F18) showing a semicomplete pennaceous feather. Scale bar, 5 mm. **b** Detail of the nymphal tick in dorsal view and barbs (inset in **a**). Scale bar, 1 mm. **c** Detail of the tick's capitulum (mouthparts), showing palpi and hypostome with teeth (arrow). Scale bar, 0.1 mm. **d** Detail of a barb. Scale bar, 0.2 mm. **e** Drawing of the tick in dorsal view indicating the point of entanglement. Scale bar, 0.2 mm. **f** Detached barbule pennulum showing hooklets on one of its sides (arrow in **a** indicates its location but in the opposite side of the amber piece). Scale bar, 0.2 mm

**Description**. See Supplementary Note 2 for body measurements.

*Male*: Body outline subcircular. Integument surface with closely spaced, deep pits and with single, mound-like elevations between pits (Figs. 3d, k, 5a–c, e), as in females (Fig. 4c). Integument not convoluted (cf. *Nuttalliella*), lacking microsculpture (e.g., granulations). Body without conspicuous setal vestiture, except setae present on palpi, legs and anal valves, and very sparse setae present on dorsal and ventral integument. Integumentary pits lacking any associated setae.

*Dorsum*. Pseudoscutum distinct (not highly chitinised as in Ixodidae, with integument resembling the rest of body), occupying most part of dorsum, reaching anterior margin of dorsum (Fig. 3d), with anterolateral margin broadened posteriorly (Fig. 5b). Cervical grooves present, relatively shallow (Fig. 5a,

b). Pseudoscutum integument with closely spaced, deep pits, but without mound-like elevations as in the rest of body, rendering a surface with smooth appearance in which pits are very apparent (Fig. 3g). Pits separated by a length equal to their diameter or less. Festoons absent. Eyes absent.

*Venter*. Capitulum partially visible in dorsal view. Hypostome subterminal (sensu Mans et al.[16]) (Figs. 3a, d, f, 5b), well developed, reaching apex of palpomere II. Hypostome ultrastructure obscure, dental formula indeterminate. Chelicerae only partially visible in the paratype male. Palpi elongated, gracile (around two times the length of hypostome), fully mobile (Fig. 4a; Supplementary Fig. 2b–d), as in females (Fig. 4b). Palpomere I short. Palpomere II the longest, distally thickened in width and height, bent distally in ventral direction (creating a ventral

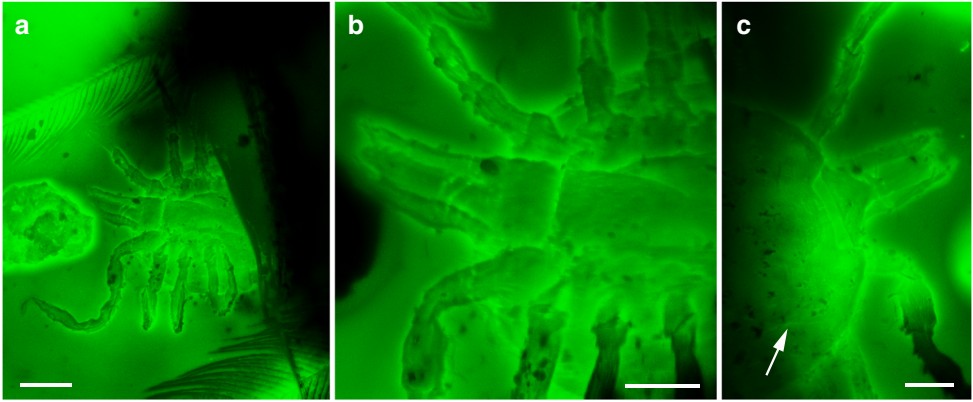

**Fig. 2** Confocal laser scanning microscopy images showing the hard tick morphology. **a** Habitus in ventral view of the *Cornupalpatum burmanicum* nymph associated with feathers. Scale bar, 0.2 mm. **b** Detail of the gnathosoma and coxal area in ventral view revealing the absence of genital pore. Scale bar, 0.1 mm. **c** Dorsal view detail of the gnathosoma and anterior part of the scutum (arrow indicates the lateral margin of the scutum). Scale bar, 0.1 mm

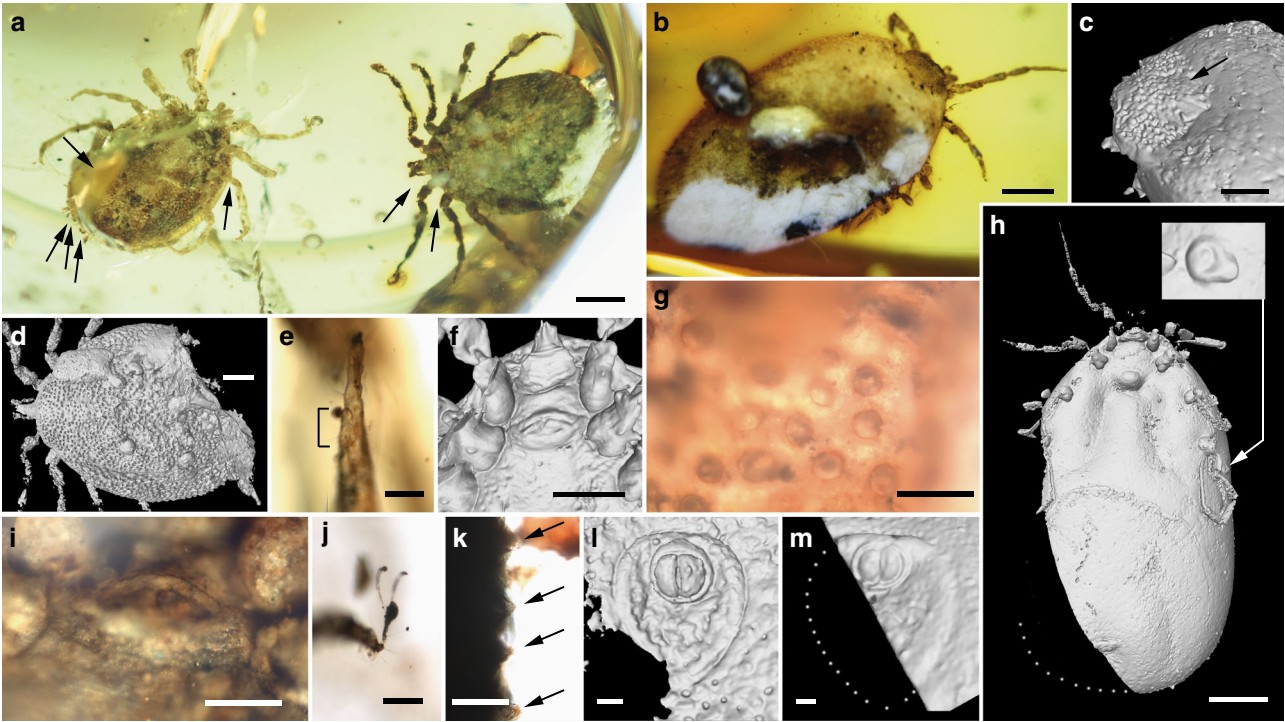

**Fig. 3** Morphology of the new tick family Deinocrotonidae. **a** Holotype (left) and paratype male in ventral view (arrows indicate the location of some entangled hastisetae of the beetle family Dermestidae). Scale bar, 1 mm. **b** Engorged paratype female in dorsolateral view. Scale bar, 1 mm. **c** Pseudoscutum (arrow) of specimen in **b**. Scale bar, 0.5 mm. **d** Paratype male in dorsal view. Scale bar, 0.5 mm. **e** Dorsal surface of the tarsus I from the holotype, showing Haller's organ, an aggregate of chemoreceptors, mechanoreceptors, and hygroreceptors in ticks for locating hosts and mates (lines mark the length of the organ). Scale bar, 0.1 mm. **f** Transverse genital aperture between coxae II, coxal spurs, and basis capituli from the holotype. Scale bar, 0.5 mm. **g** Pitted dorsal integument without elevations in the pseudoscutum of the same specimen. Scale bar, 0.1 mm. **h** Engorged paratype female in ventral view with detail of the spiracle. Scale bar, 1 mm. **i** Genital aperture between coxae II of the paratype male. Scale bar, 0.2 mm. **j** Pulvillus and pretarsal claws of the holotype. Scale bar, 0.1 mm. **k** Lateral body margin showing the non-convoluted, mound-like elevations of the integument (arrows) between pits of the same specimen. Scale bar, 0.1 mm. **l, m** Anus and preanal groove of the paratype male and engorged paratype female, respectively. Scale bars, 0.1 mm. **a, b, e, g, i–k** obtained with compound microscopy, the remainder with CT-scans

concavity, with surface of articulation with palpomere III facing that direction). Palpomeres III and IV elongated, tubular, tapering basally. Palpomere III about two times as long as wide, with surface slightly ruffled. Palpomere IV in terminal position, about four times as long as wide. Palpi without spurs but bearing abundant, fine setae. Basis capituli not bordered by coxae I, with anterior margin rimmed and surface smooth (Fig. 3f; Supplementary Fig. 2b); auriculae, cornua and porose areas absent.

Genital aperture a transverse slit in an oval area between anterior half of coxae II (Fig. 3f, i), close to capitulum. Presence of a conspicuous anteroventral depressed area (Fig. 5g) that is quadrangular in shape and post-genital, laterally limited by anterior section of genital groove. Genital groove well developed and extending posteriorly; medially divided (immediately after coxae IV) into two sections (Fig. 5g). Anterior genital groove section extending from coxae II to IV, briefly bordering coxae IV

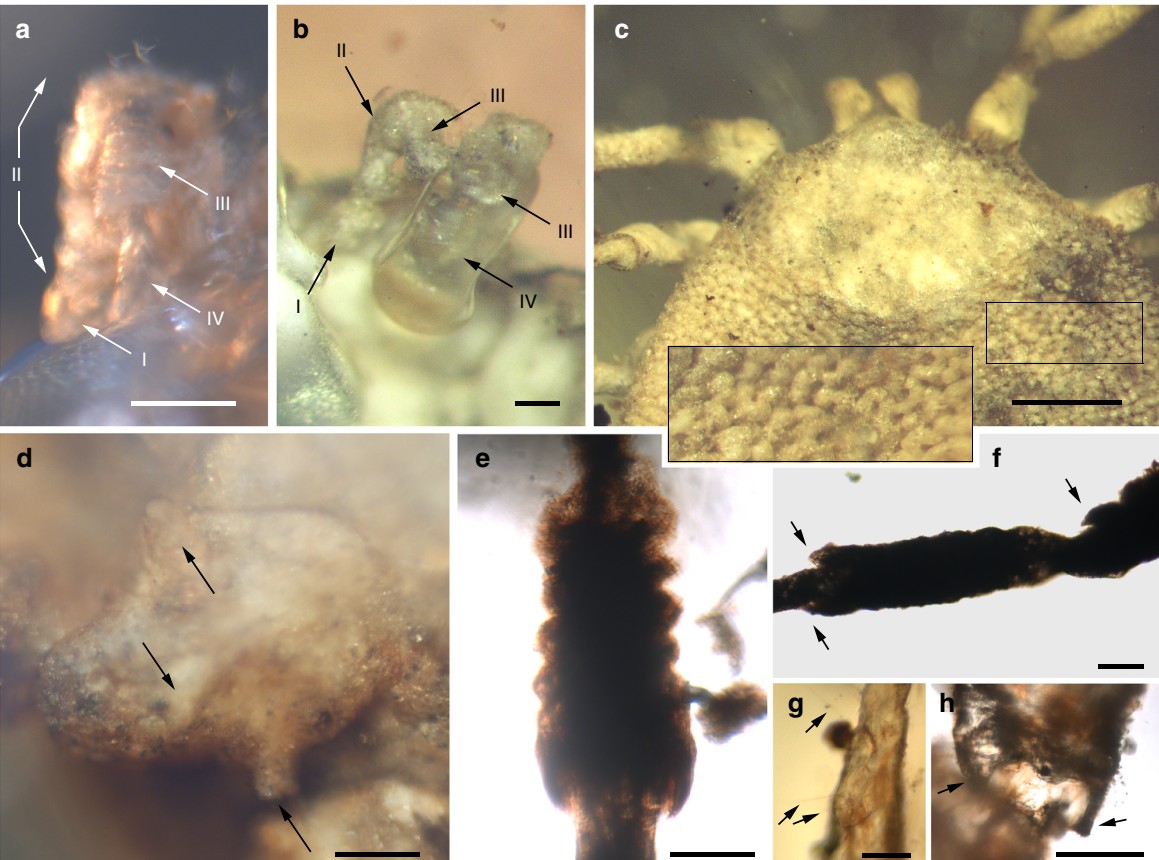

**Fig. 4** Photomicrographs showing some anatomical features of the new family Deinocrotonidae. Holotype (AMNH Bu-SA5a) (**a**, **f**, **h**); allotype (CM 63007) (**b**, **c**); paratype male (AMNH Bu-SA5b) (**d**, **e**). **a–b** Right palp and right and left palpi in ventral views, respectively, with indication of the number of visible palpomeres. Scale bars, 0.1 mm. **c** Pseudoscutum and detail of the integument showing mound-like elevations between the pits (see inset). Scale bar, 0.5 mm. **d** Coxa II showing a row of three spurs (arrows). Scale bar, 0.1 mm. **e** Ruffled surface of the left genu III. Scale bar, 0.1 mm. **f** Articulations of the left leg III in ventral view. Note the notch-like processes (arrows). Scale bar, 0.1 mm. **g** Haller's organ in dorsal surface of the tarsus I (bottom structure is the proximal capsule, in contact with the distal pit). Arrows point to sensilla. Scale bar, 0.05 mm. **h** Trochanterofemoral articulation of the right leg I. Note the notch-like processes (arrows). Scale bar, 0.1 mm

distally (i.e., diverging towards body margin). Posterior genital groove section the longest, grooves progressively diverging posteriorly, slightly bordering anal plate. Spiracle well developed and very close to body margin at level of coxae IV, smaller than in Ixodidae (and in a different position) and larger than in *Nuttalliella namaqua*[17]. Spiracle plate structure sub-triangular in shape and consisting of a small macula and a smooth triangular plate, not fenestrated but bearing two small concavities (Fig. 5e, f), as in females (Figs. 3h, 5h); macula projecting towards ostium to form a lip; entire plate arising from a depressed cuticular area. Preanal groove prolonged posteriorly, with sides closing, delimiting a guitar pick-shaped anal plate (Fig. 3l), as in females (Fig. 3m). Anal pore close to posterior margin of body. Anal valves with a few long and fine setae.

Legs. Long and strongly flattened laterally from trochanters to tarsi; arising within anterior two-fifths of total body length. Leg joints not of ball and socket type as in *Nuttalliella*, but leg articles with paired, notch-like ventrodistal processes (without forming sockets for the articulation, balls not distinct), more apparent in basal articulations (Figs. 4f, h, 5c, d). Slight separation between coxae, except coxa I contiguous with II. Coxae armed with rows of small, shallow spurs (i.e., rounded tubercles, such as in some ixodids and *Nuttalliella*) (Figs. 3f, 4d, 5f): one spur on coxa I—in medioposterior position—and three on each coxa II, III, and IV. Three coxal spurs forming a row in coxa II, with two of them in a medial, posterior position while third one in a distal, anterior

position. Three coxal spurs aligned in medial position in coxae III and IV (two close together in a slightly basal, posterior position and third one in anterior position at middle of coxa). Trochanter without spurs. Femur, genu, and tibia bearing a sculptured surface of transverse ridges (ruffles), especially marked in genu (Figs. 4e, 5d). Trochanters I and II with very shallow ruffles, almost indistinct. First pair of legs with deeper ruffles. Femora I and II positioned very high and strongly flattened laterally. Femur III flattened laterally and high only basally. Femur IV tubular. Haller's organ conspicuous; although only observed in right tarsus I of holotype (Figs. 3e, 4g; Supplementary Fig. 2e) due to preservation of remaining specimens, situated on a dorsal elevation of tarsus I and composed of two parts, a completely open (without a transverse slit) proximal capsule having long setae and a distal pit followed by more long, distinct setae, capsule larger than pit. Basitarsus as long as tarsus in legs II–IV. Pretarsi with two curved pretarsal claws and abundant, long setae. Pretarsal claws large. Pulvilli poorly developed (Fig. 3j).

*Female*: As in male with the following exceptions: Integument, including that of pseudoscutum, with pits not as well defined as in males. Pseudoscutum abbreviated (Figs. 3c, 4c; Supplementary Fig. 3), occupying the anteriormost part of dorsum. Genital aperture in a more posterior position than in males, between coxae II and III, and apparently showing a smooth surface (Supplementary Fig. 3). Marginal groove absent.

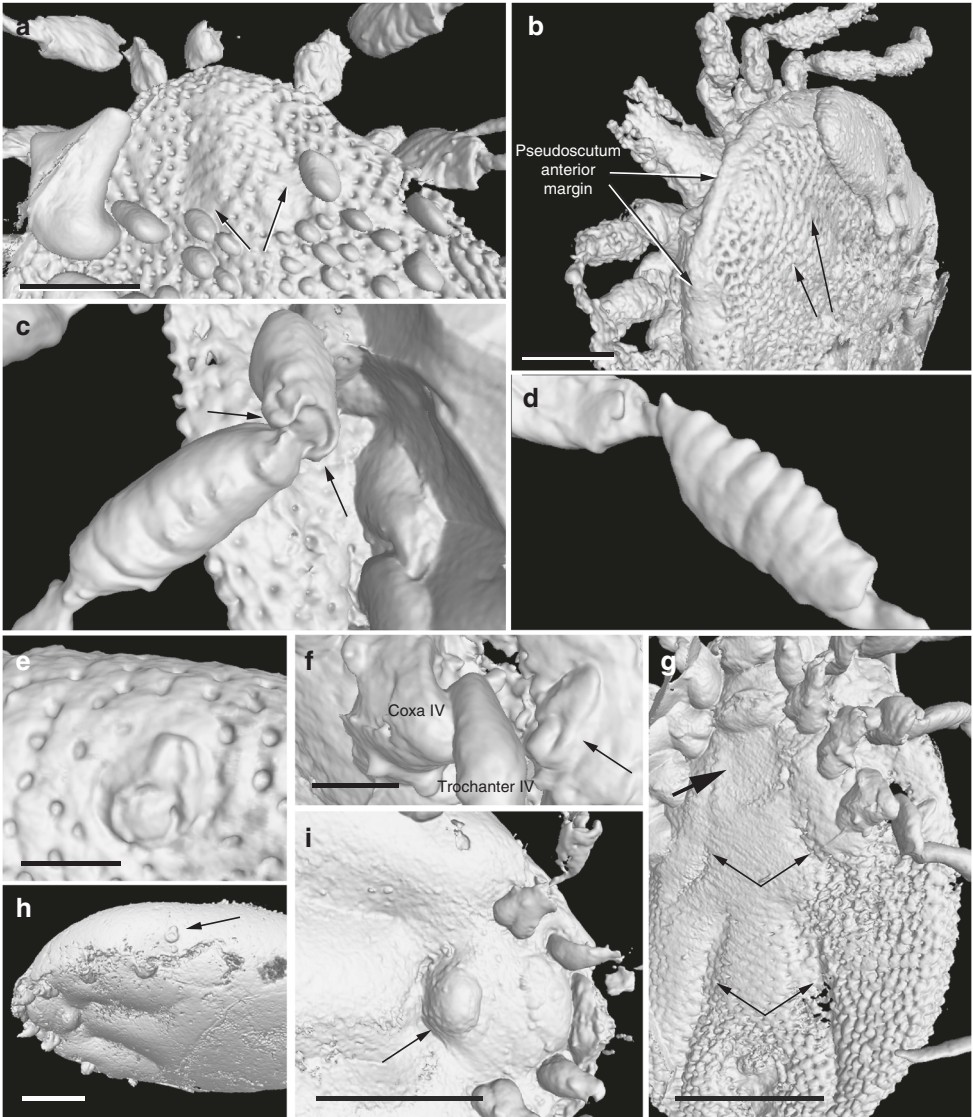

**Fig. 5** CT-scan images showing some anatomical features of the new family Deinocrotonidae. Holotype (AMNH Bu-SA5a) (**a**, **c**, **d**, **f**, **g**); paratype male (AMNH Bu-SA5b) (**b**, **e**); engorged paratype female (CM 63001) (**h**, **i**). **a** Pseudoscutum showing the cervical grooves (arrows). Note the abundant bubbles (bottom). Scale bar, 0.5 mm. **b** Pseudoscutum in anterodorsal view showing its posteriorly broadened anterior margin and the cervical grooves (right arrows). Scale bar, 0.5 mm. **c** Trochanterofemoral articulation of the right leg III (femur length ca. 0.5 mm). Note the notch-like processes (arrows). **d** Ruffled genual surface (genu length ca. 0.6 mm). **e** Right spiracle in frontal view. Scale bar, 0.2 mm. **f** Left spiracle in lateral view (arrow). Scale bar, 0.2 mm. **g** Post-genital, anteroventral depressed area (bold arrow) and genital groove medially divided in two sections (thin arrows). Scale bar, 1 mm. **h** Habitus showing the deformation of the body and the completely stretched integument due to engorgement (arrow indicates the spiracle). Scale bar, 1 mm. **i** Detail of the ventral surface showing the genital aperture extruded as a rounded protuberance (arrow). Scale bar, 1 mm

**Remarks**. A suite of unique, presumably derived characters defines Deinocrotonidae: the integument structure, the palp morphology, and the shape of the preanal groove. Likewise, the discontinuous genital groove is unique among ticks. The subterminal hypostome and the presence of a pseudoscutum suggest a close relationship between Deinocrotonidae and Nuttalliellidae. Pending a phylogenetic analysis when more material is available (see Supplementary Note 3), we propose here that both families are sister to (Ixodida + Argasidae). So far, a few more deinocrotonids have been found in Burmese amber, and one additional undescribed immature specimen from 105 Ma old Spanish amber most likely belongs to this new family. Apart from the unique characters among ticks, the new family differs from Nuttalliellidae in the following features (see Supplementary Tables 1 and 2): (1) pseudoscutum pitted (vs. mesh-like), (2) pseudoscutum reaching the anterior margin of the dorsum in males, (3) cervical grooves present, (4) capitulum not bordered laterally by coxae I, (5) basis capituli simple and with smooth surface, (6) cornua absent, (7) genital area smooth (vs. irregularly striated), (8) anteroventral depressed area in post-genital position (vs. in pre-genital position), (9) all coxae armed and spurs forming rows, (10) leg joints not of the ball and socket type, at least as in *Nuttalliella*, (11) proximal capsule of Haller's organ completely open, (12) different morphology and size of the spiracle, and (13) preanal groove different in microscopic detail (smooth vs. posterior and anterior margins with dentate integumental projections).

The pseudoscutum in Deinocrotonidae occupies most of the dorsum in males and is abbreviated in females, as occurs in ticks with a scutum/pseudoscutum. The special shape of palpomere II, distally thickened and bending distally in a ventral direction (Fig. 4a, b; Supplementary Fig. 2b–d), appears to be an adaptation to protect the distal part of the gnathosoma dorsally and

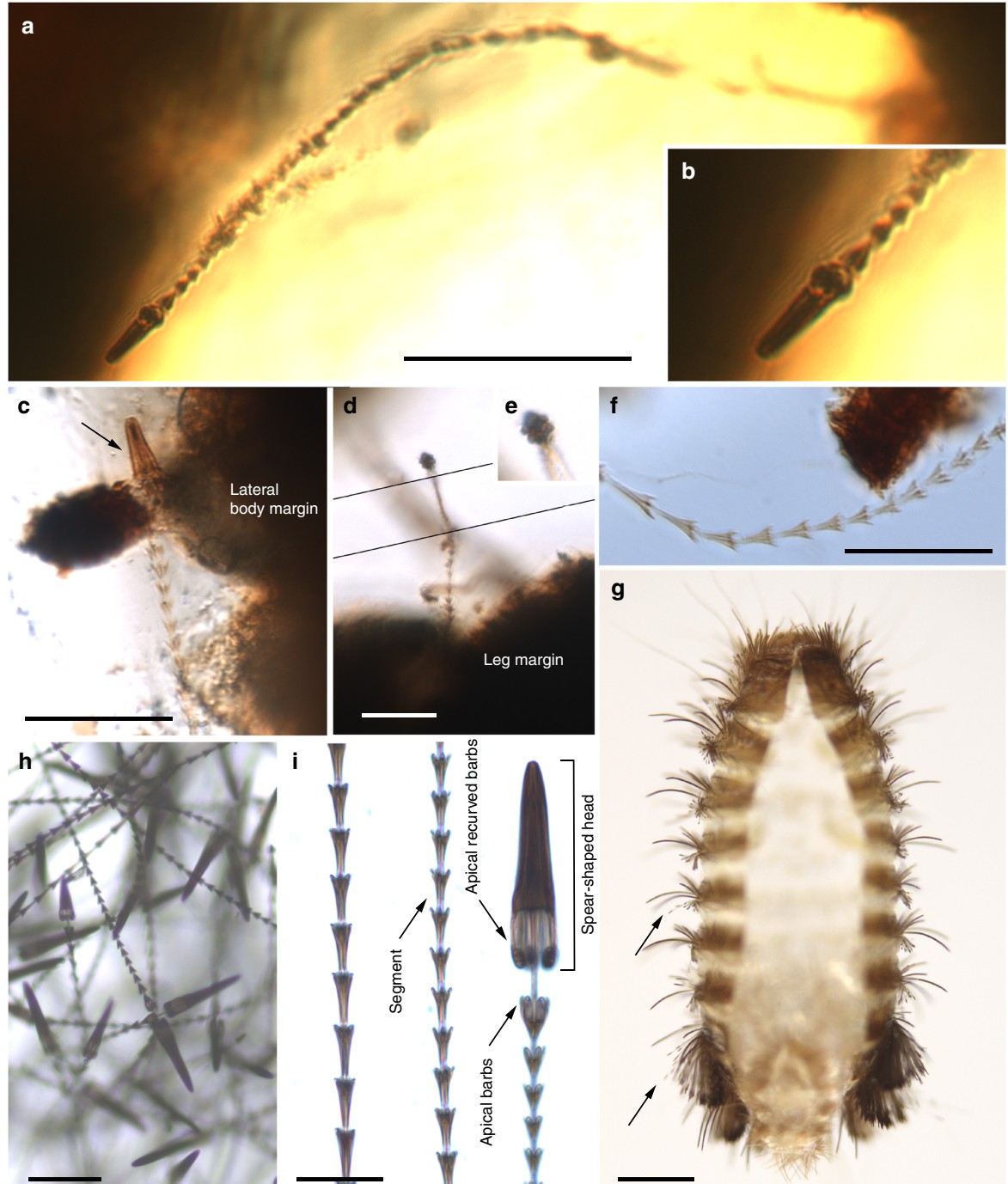

**Fig. 6** Hastisetae on the two deinocrotonid ticks preserved together and comparisons with extant Megatominae. **a** Hastiseta preserved with its spear-shaped head entangled in a leg of the paratype male (AMNH Bu-SA5b). Scale bar, 0.1 mm. **b** Detail of the spear-shaped head of the hastiseta from **a**. **c** Hastiseta with the spear-shaped head (arrow) entangled in the holotype (AMNH Bu-SA5a). Scale bar, 0.05 mm. **d** Hastiseta with the spear-shaped head photographed from above entangled in the base of the right femur I of the paratype male. Scale bar, 0.05 mm. **e** Spear-shaped head magnified from **d** showing its six knobs. **f** Multi-segmented portion of a hastiseta, without preserved head, on the posterior body margin of the holotype (segments to the right are distal). Scale bar, 0.05 mm. **g** Extant larval cast-off skin after molt in dorsal view of the Megatominae genus *Anthrenus* (arrows indicate two of the hastisetal tufts on abdominal segments), which can be found in bird nests. Scale bar, 0.5 mm. **h** Several hastisetae from a posterior tuft from **g**. Scale bar, 0.05 mm. **i** Basal (left), middle and distal (right) multi-segmented sections of one hastiseta from **h**. Scale bar, 0.02 mm

anteriorly, especially the delicate teeth of the hypostome and the chelicerae. Such expansion of the distal part of the palpomere II is present in all ixodids (namely their upper inner margin, creating an inner groove), although palpomere III is also expanded, taking part in the protection of the gnathosoma, and both palpomeres are straight, directed forwards[18,19]. In *Deinocroton*, palpomere III

is elongated and tubular, directed ventrally due to the surface of articulation between palpomeres II and III facing that direction and due to the shape of the palpomere II. In *Nuttalliella*, palpomere II is massive, expanded laterally and provides most of the gnathosomal protection; palpomere III is smaller, triangular in shape and slightly laterally expanded ventrally, whereas both

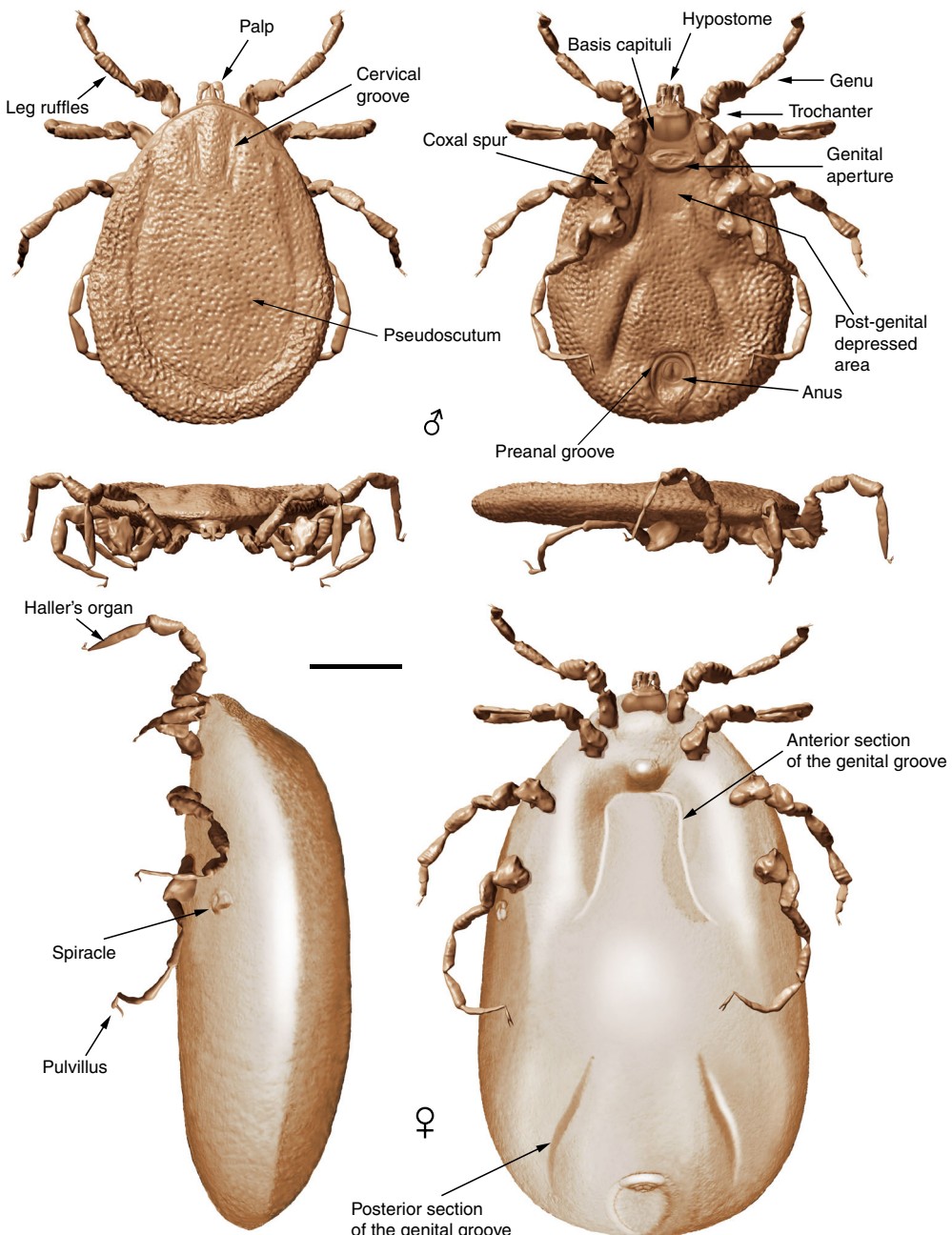

**Fig. 7** Reconstruction of the male and engorged female of *Deinocroton draculi*. Upper dorsal, ventral, frontal, and lateral views based on CT-scans of the holotype male (see Supplementary Movie 1) (Artist: Oscar Sanisidro). Lower lateral and ventral reconstructions based on CT-scans of the engorged paratype female (performed by the authors using elements from the male model performed by O. Sanisidro). Both reconstructions at the same scale and with modifications based on compound microscope observations. Scale bar, 1 mm

palpomeres are straight, directed forwards as in ixodids[20]. Argasids lack any palpomere expansion for gnathosomal protection due to the ventral position of their capitulum in adults. On the other hand, the Haller's organ in deinocrotonids has a generalised morphology, with a proximal capsule and a distal small pit, but fine details are obscure under optical microscopy and they have remained unresolved using CT-scanning. Nevertheless, the proximal capsule is fully open (lacking a transverse slit) as in *Ixodes*, and unlike in other ixodids, argasids and *Nuttalliella*[21–23]. Furthermore, CT-scanning revealed the spiracular morphology and position in detail, which are very similar to those of Argasidae[24]. Although the spiracle position in *Deinocroton* is coincident with that of *Nuttalliella*, the latter has a

minute spiracle with a cribose spiracular plate[20]. Also, the spiracle of the new family is quite different from that in Ixodidae (i.e., bigger and in a posterior position, not hidden by coxae IV[19, 25]). Lastly, the ventroposterior grooves that are posterior to coxae IV and diverge towards the posterior body margin have been named herein "posterior genital groove sections", despite not being connected to the longitudinal grooves that arise from the genital area. Although the origin of these posterior grooves is unclear, the set of the anterior and posterior sections is very similar in position and extension to the genital grooves of some ixodids. Other ixodids, such as *Boophilus*, have posterior grooves due to the presence of adanal shields; however, since *Deinocroton* lacks any structure resembling this shield, the posterior section of the

genital groove in the new family appears to be unique among ticks.

The holotype and paratype male *Deinocroton*, preserved together, have at least seven spear-headed, multi-segmented setae of exogenous origin attached to their bodies (Fig. 6; Supplementary Fig. 4). The longer setae remains are 311 μm (Fig. 6a; Supplementary Fig. 4b) and 286 μm (Fig. 6f; Supplementary Fig. 4e) in length as preserved and contain 27 segments plus its spear-head and 23 segments, respectively. The spear-head is 27 μm long, 5 μm wide (11 μm in the base), more sclerotised than the rest of the seta and with six basal knobs arranged in circle. The basalmost segments are long (23 μm long the longest preserved) and quickly decrease in length towards the apex of the seta. The distal setal section shows short segments of similar length (ca. 9 μm long in the 20 distal segments), with the distalmost segment (that in connection with the spear-head) not differing in shape and size from the immediately preceding ones.

Despite the dilated body of the engorged specimen (paratype female), it belongs to *Deinocroton draculi* based on the virtually identical size and morphology of the capitulum (including the basis capituli), pseudoscutum, legs (including the relative length of leg segments), two sections of the genital groove, spiracle and anal plate. The morphoanatomical changes in the engorged specimen when compared to the three unengorged ones (attributed to engorgement) are as follows (Figs. 3b, c, h, m, 5h, i, 7; Supplementary Table 3): (1) the body increased ca. 1.7 times its length, ca. 1.4 times its greatest width, and ca. 3.6 times its greatest height—this corresponds to an approximate volume change from 15.0 to 126.0 mm³ (i.e., a volume increase of ca. 8.5 times); (2) the dorso-ventrally planar body became inflated (more pronouncedly so medially along the longitudinal axis) and its subcircular outline became elongated (bean-shaped), particularly in the transverse medial portion of the body or area that separates the anterior and posterior sections of the genital groove; (3) the body integument became smooth, without evidence of the original pits; (4) the post-genital depressed, quadrangular area disappeared; (5) coxae became strongly separated from one another, particularly coxae II from III and III from IV; (6) the genital aperture became deformed to a plate with a globular extruded protrusion; (7) the spiracle was displaced to a posterior position regarding coxae IV, but without changes in its morphology and size; and (8) the anal plate became dilated (its greatest width increased by one-and-half times) but the anal valves remained unchanged in morphology and size. It is noteworthy that the pseudoscutum preserved its size and pits in the engorged specimen, without signs of dilation, as in the allotype. The pseudoscutum does not change its morphology with engorgement in *Nuttalliella* either[5]. The engorged *Deinocroton* represents the third engorged tick known in the fossil record; the other records have been found in Cretaceous Burmese amber[7] and Miocene Dominican amber[26].

## Discussion

The relatively loosely vaned pennaceous feather that the hard tick described herein is grasping (Fig. 1; Supplementary Fig. 1) shows barbule pennula with hooklets in some areas. This would assign the feather to stage IV in Prum's evolutionary-developmental model of the feather, but the clear length asymmetry between the barbs on either side of the rachis classifies it within stage V[27]. Even though stage IV and V feathers have for the most part been inferred in the fossil record, namely in compression fossils through the presence of well-developed closed vanes, some directly visible instances of these stages in Cretaceous amber feathers were previously reported (although not figured or poorly so) bearing barbules with hooklets like the ones presented here[28],

[29]. These structures have not been described from other Cretaceous feathers found in Burmese[30, 31], Canadian[32], or Spanish ambers. Furthermore, stage IV feathers have been associated with taxa adapted for gliding or powered flight due to the ability of the barbules to interlock and allow for closed feather vanes[27], but as the latter are also found in cursorial taxa they do not directly imply gliding or flying ability[33]. In any case, a feather belonging to the stage V indicates that the dinosaur host of the hard tick described herein falls within the clade Pennaraptora according to current evidence from the fossil record of feathered dinosaurs (see Supplementary Note 4). Crown-group birds are excluded as possible hosts because their inferred age is significantly younger than Burmese amber, i.e., about 73 Ma based on targeted next-generation DNA sequencing[34]. Even if the palpal claws of *Cornupalpatum* were interpreted as a possible adaptation to parasitism of an extinct line of reptilian hosts[12], at least the nymphs ectoparasitised feathered dinosaurs based on the direct evidence provided herein, although this hard tick species could have also parasitised other hosts.

The tick is entangled with the feather's barb in virtually the same orientation, indicating that both contacted the resin together after separation from the feathered host. Such contact occurred at the ground level as indicated by the overall fossil assemblage preserved in the amber piece (see Supplementary Note 1), although both the feather carrying the tick and/or the resin that encased them could have fallen from above. Entrapment within different resin flows of the feather and the tick is implausible because the resin is a viscous medium in which the entanglement of both entities cannot occur by slow contact due to drift into that medium.

The seven spear-headed setae attached to the two *Deinocroton* preserved together have a unique morphology that occurs in some larvae of the beetle family Dermestidae. Larval dermestids have a body vestiture that usually includes one or more types of modified setae bearing spicules and/or recurved hooks. Among them, hastisetae (multi-segmented, spear-headed setae) with apical recurved barbs are found in the subfamily Megatominae and some Trinodinae: Trinodini[35]. These specialised setae form conspicuous tufts on some abdominal tergites, namely on the posterior abdominal segments (Fig. 6g). Hastisetae become easily detached and serve as a defensive mechanism by sticking to the appendages of potential predators and entangling them or at least hampering mouthpart activity while the larvae have enough time to escape[35–37]. In extant large dermestid populations, such as those occurring in nests, detached hastisetae and larval exuviae can form hastisetal mats[36]. The structure of the hastisetae entangled on the two *Deinocroton* (Fig. 6; Supplementary Fig. 4), namely the well-defined whorls of apical barbs on the hastisetal segments and the conspicuous knob-like recurved barbs basally on the spear-head, shows that they are most likely affiliated with Megatominae[35, 37]. Dermestid larvae, and megatomines in particular, are often found in nests in a commensalistic relationship with the nest-producer, feeding on shed feathers and other organic detritus[35] (see Supplementary Note 5). Indeed, bird nests are seasonally rich sources of organic material in a sheltered micro-environment that sustains a wide diversity of beetles, moths, mites, fleas, and other arthropods[38]. Nesting behaviour has been substantially proven in non-avialan theropods and other Mesozoic dinosaurs[39].

The two *Deinocroton* preserved together are very close to each other in the amber piece and have the same dorso-ventral orientation, indicating that they contacted the resin surface in a similar fashion, and thus at nearly the same time. The exceptional co-occurrence of two ectoparasites in an amber piece can be most parsimoniously explained by the new species being nest-inhabiting (nidicolous), so that the two specimens belonged to

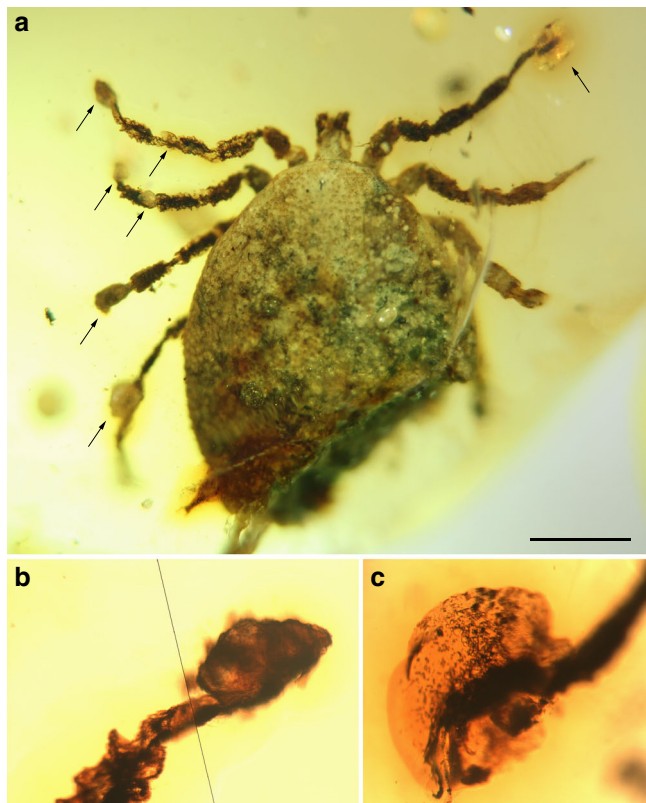

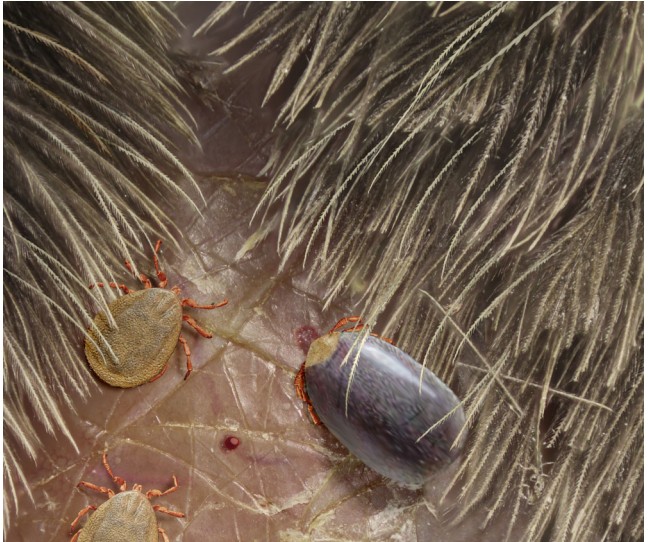

**Fig. 9** Reconstruction of the habitus of *Deinocroton draculi* on an immature feathered dinosaur. The reconstruction shows two unengorged males (left) and a female feeding to engorgement (right). Male body length ca. 3.9 mm. Colours of the ticks are conjectural but based on the colouration seen in the related nuttalliellid ticks. Performed by the authors using models of the males created by the artist Oscar Sanisidro

**Fig. 8** Amber drops attached to legs of the deinocrotonid paratype male (AMNH Bu-SA5b). **a** Dorsal view of the specimen (arrows indicate the amber drops). Scale bar, 1 mm. **b** Left tarsus I dorsally covered by an amber drop. Scale bar, 0.2 mm. **c** Right tarsus I coated by an amber drop with abundant bubbles inside (arrow indicates the claws). Scale bar, 0.2 mm

the same tick population. Extant nidicolous ticks live in the host's nest or in a nearby harbourage, as opposed to non-nidicolous ticks, which seek hosts in the open environment (=questing)[18]. Whereas non-nidicolous ticks tend to have more dispersed individuals, nidicolous ticks can aggregate in high numbers at their nesting areas[18, 24]. From an evolutionary standpoint, nidicoly is thought to have been an ecological precursor for parasitic relationships to develop, first transitioning from temporary to permanent nidicoly; then from saprophagy to feeding on excretory products or shed/sloughed remains from vertebrates; then to feeding on skin, integumentary structures (such as feathers), secretions and blood from wounds; and lastly developing structures to damage the host's skin[40]. Although it is unclear if *Deinocroton draculi* inhabited its host's nest or lived in its own nest nearby that of the host, the presence of hastisetae on the two *Deinocroton* preserved together indicates that most likely the ticks had been in the host's nest immediately before becoming entrapped in resin. Indeed, if both ticks had become entombed in a resin emission while questing among the vegetation or following contact with their common host, the presence of dermestid hastisetae on both specimens would be less likely. Also, the unengorged status of both *Deinocroton* makes the latter scenario even more improbable—unless extraordinary circumstances occur, ticks voluntarily detach from the host only when a feeding cycle is completed and conditions are favourable, a behaviour termed "dropping"[18, 24]. Lastly, the absence of the host's integumentary remains (e.g., feathers) in the amber piece casts further doubt on the idea that the two ticks became entrapped in

resin directly by incidental contact of their host to a resin emission.

The presence of dermestid hastisetae in the two *Deinocroton* preserved together and the inferred nidicolous ecology of these ticks, when considering the scarce record of hairs[41] vs. that of feathers[28–31] in Cretaceous amber (particularly in Burmese amber), allows us to infer that deinocrotonids most likely included feathered dinosaurs among their hosts. In addition, the paratype male of *Deinocroton* has seven amber drops stuck to some of its leg apices (Fig. 8a), namely those on the left side. These amber drops, abundantly reported from *Proplebeia* bees in Dominican amber[42], are distinct from the surrounding amber matrix due to their darker colour and bubble content (Fig. 8c), and indicate that, before becoming entombed in amber, the tick likely first made contact with fresh resin but managed to escape. During that event, however, the tick's two Haller's organs became completely coated with resin (Fig. 8b, c), and thus the capacity of the tick to detect hosts using these aggregates of receptors was severely hindered; also, the tick partially lost its ability to attach to hosts as the claws and pulvilli of both its first legs were rendered impaired as well. After the resin drops on the tick's legs rounded and hardened, the specimen became embedded in resin near another conspecific tick (holotype) of the same developmental stage and feeding status. Therefore, a tick with reduced capacity to detect and attach to hosts further undermines the idea that the two *Deinocrotonids* preserved together had recently been in contact with their feathered host or had detached from it. The two ticks were most likely caught by resin nearby the feathered host's nest, where the dermestid hastisetae became attached to their bodies.

The engorged *Deinocroton* specimen shows morphoanatomical changes indicating full blood engorgement, such as a completely dilated integument (without pits) and an extruded genital area (Fig. 7). This indicates that this particular specimen contacted a resin flow soon after it dropped from its host, once it had completed its blood meal. The pitted, highly extensible integument of deinocrotonids, and a body volume increase of ca. 8.5 times when engorged in females, suggest that their adult feeding was like that

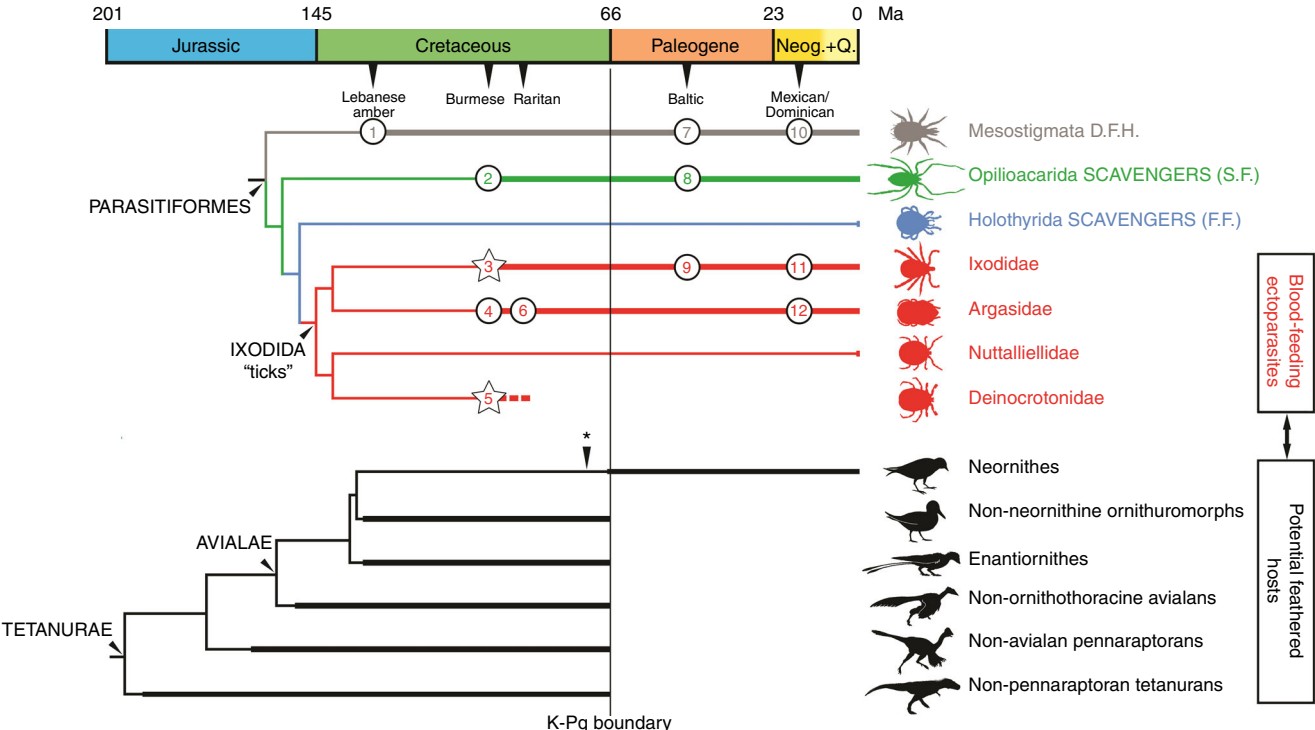

**Fig. 10** Ticks and their possible feathered hosts in deep time. Simplified phylogenies of parasitiform Acari (top) and tetanuran Dinosauria leading to the bird lineage (bottom). Although filamentous integumentary structures are known in some ornithischians and pterosaurs, the latter are not represented for not belonging to the bird lineage. Time ranges supported by the fossil record are depicted with thick lines; those inferred appear in thin lines. Asterisk marks the inferred origin of modern birds (Neornithes). Known fossil occurrences of parasitiform mites (all in amber; stars correspond to tick records that can be related to feathered dinosaur hosts, presented in this paper; quaternary records excluded): Lebanese amber—1, Mesostigmata indet.; Burmese amber—2, ? *Opilioacarus groehni*; 3 *Cornupalpatum burmanicum*, three specimens including the new one described herein entangled in a pennaceous feather; *Compluriscutata vetulum*, *Amblyomma birmitum*, and *Amblyomma* sp.; 4 Argasidae indet.; 5 *Deinocroton dracuii* (herein); Raritan amber—6, *Carios jerseyi*; Baltic amber—7, *Sejus bdelloides*; *Aclerogamasus stenocornis*; Microgynioidea indet.; 8 *Paracarus pristinus*; ?*Opilioacarus aenigmus*; 9 *Ixodes succineus* and *Ixodes* sp.; Mexican amber—10, *Dendrolaelaps fossilis*; Dominican amber—11, *Amblyomma* sp.; 12 *Ornithodoros antiquus*. An unpublished, badly preserved specimen from Spanish amber (105 Ma) has been not included, but it could be assignable to Deinocrotonidae. See Supplementary Notes 3, 4, and 6 for inferred time ranges used, parasitiform records shown, oldest occurrences of dinosaur groups depicted, and discussion on feather evidence in non-avialan dinosaurs. Neog. + Q. Neogene and Quaternary, D.F.H. diverse feeding habits, S.F. solid feeders, F.F. fluid feeders

observed in nuttalliellids and soft ticks, feeding rapidly to engorgement (in minutes to hours) and having multiple gonotrophic cycles[5, 18, 43]. This strategy contrasts with that of hard ticks (see Supplementary Table 3), since their adult females can increase more than a hundred times their original body volume through active growing of the cuticle while they feed, which occurs only once and for periods that can last weeks[5, 18, 43]. It has been hypothesised that arthropods such as dipterans and several extinct flea-like groups were vectors of disease in dinosaurs and pterosaurs[44–46]. The ticks described herein are additional candidates for disease vectors of feathered dinosaurs. Microscopic structures putatively resembling the size and shape of rickettsial proteobacteria have been described from the midgut of *Cornupalpatum burmanicum*[47], although these must be independently examined. Since hard and soft ticks are today vectors of disease among birds, mammals, and reptiles[18, 43], and *Nuttalliella namaqua* also parasitizes these hosts[48], deinocrotonids likewise could have spread diseases among the Mesozoic relatives of these vertebrates.

Direct evidence herein proves that hard ticks fed on blood from feathered theropods (non-avialan or avialan) during the latest Early Cretaceous, showing that the parasitic relationship that today binds ticks to birds was already established among early representatives of both lineages and has persisted for at least 99 million years. Deinocrotonids most likely were ectoparasitic on feathered dinosaurs (Fig. 9) as well based on the sum of evidence

presented above. Most of the hematophagic and ectoparasitic strategies in insects developed during the Mesozoic[44], and such adaptations extend back to at least that time for ticks as well, and likely earlier. Some flea-like extinct insect groups that are considered potential haematophagous ectoparasites of vertebrates did not survive to the Cainozoic[46, 49]; all are known as isolated compression fossils and lack direct evidence of their hosts. Similarly, and unlike the remaining tick lineages, deinocrotonids went extinct during the Cretaceous, possibly at the K–Pg extinction event, together with their feathered dinosaur hosts if these ticks were host specialists (Fig. 10). In any case, host specificity is not considered an important factor in the evolution of ticks[50], and many extant species are generalists, including *Nuttalliella namaqua*[48]. Further direct evidence of early ectoparasitic hematophagy must be sought in rich Cretaceous ambers, or by carefully screening the preserved feathered or haired vestitures of vertebrates from Jurassic and Cretaceous Konservat-Lagerstätten, such as the rich compression deposits from China.

## Methods

**Material**. The specimen Bu JZC-F18 was donated to the American Museum of Natural History (AMNH) by James Zigras where it is housed within the Zigras Collection. The specimens AMNH Bu-SA5a/b, CM 63001, and CM 63007 (Hukawng Valley, Myanmar) are included in three polished pieces of Burmese amber, and are housed at the AMNH and the Carnegie Museum of Natural History, respectively, by donation of one of the authors (S.R.A.). The preparation of

some previously polished amber pieces was improved by trimming the amber surfaces using a scalpel and re-polishing.

The amber piece containing the ixodid *Cornupalpatum burmanicum* (Bu JZC-F18) is transparent, light yellow, and $4.7 \times 1.8 \times 0.7$ cm in size. The amber piece containing the deinocrotonid holotype (male) and paratype male (AMNH Bu-SA5a and AMNH Bu-SA5b, respectively) is transparent, light yellow, and $1.7 \times 1.4 \times 0.4$ cm in size. The amber piece containing the allotype (female) (CM 63007) is transparent, yellow, and $1.4 \times 1.1 \times 0.4$ cm in size.

**Anatomical research and imaging.** An Olympus BX51 transmitted-light compound microscope was used to study the ticks in dorsal and ventral views. Photography of the specimens used both ColorView IIIu Soft Imaging System attached to an Olympus BX51 compound microscope and an Olympus C-5050 Zoom digital camera attached to an Olympus SZ X9 stereomicroscope. Line drawings were prepared using an Olympus U-DA drawing tube attached to the Olympus BX51 compound microscope. A Leica DM750P compound microscope with an attached Leica DFC420 camera using the software Leica Application Suite v.4.8.0 was used for pictures shown at the Supplementary Fig. 1e, f.

*Confocal microscopy*: The ixodid specimen was imaged using a Leica TCS SPE-DM 5500 CSQ V-Vis (Manheim, D-68165, Germany) at the Museo Nacional de Ciencias Naturales in Madrid (MNCN). The images were acquired with a solid-state laser operating at 488 nm, a 10× eyepiece, HCX PL FLUOTAR 5×/0.15, ACS APO 10×/0.3 dry objectives and the Leica Application Suite Advanced Fluorescence software (Leica MM AF 1.4). Fluorescence emission was collected from approximately 10 nm above the excitation wavelength up to 800 nm. Laser power for acquisition was set by viewing the fluorescence emission and increasing the power until the rate of increase in fluorescence appeared to have slowed. The photomultiplier gain for acquisition was then set by viewing the image and increasing the gain until signal overload was detected, at which point the gain was reduced slightly. Pixels matrices of $2048 \times 2048$, with speed 400 Hz, and frame average of 4 were acquired for each Z-step at a zoom setting of 1.5–3. An Airy unit setting of 1 was routinely used for the observation pinhole.

*CT-scan*: The deinocrotonid ticks were imaged at the MNCN with a Nikon XTH160 X-ray micro-CT system to obtain high-quality 3D images to complement the structures visualised using optical microscopes. Images were generated at an X-ray voltage of 64 kV (and 129 uA; voxel size 5.98 μm) for the holotype and the paratype male (AMNH Bu-SA5a and AMNH Bu-SA5b) and 80 kV (and 76 uA; voxel size 7.77 μm) for the engorged paratype female (CM 63001). Four frames per projection were acquired with an integration time of 1000 ms for a total of 1800 projections. Acquired images were rendered and visualised with VGStudio MAX 2.2 (Volume Graphics).

The 3D models of the holotype and engorged paratype female were prepared by modifying their respective CT-scan files using the software Pixologic ZBrush. Supplementary Movie 1 was created using the software Blender v.2.78.

The estimation of the volume of the unengorged and engorged tick specimens was calculated by considering their bodies as cylinders with an ovoid base. Therefore, $V = Aa \times Ab \times h \times \Pi$, where "V" is the total tick volume, "Aa" represents the longest axis, "Ab" is the shortest axis, and "h" is the height.

**Nomenclature.** Nomenclature used for the tick description follows that of Sonenshine and Roe[18], for feather description follows that of Robertson et al.[51], and for hastisetae description follows that of Lawrence and Ślipiński[35].

**Nomenclatural acts.** This published work and the nomenclatural acts it contains have been registered in ZooBank, the proposed online registration system for the International Code of Zoological Nomenclature. The ZooBank LSIDs (Life Science Identifiers) can be resolved and the associated information viewed through any standard web browser by appending the LSID to the prefix "http://zoobank.org/". The LSIDs for this publication are: DD885432-BDE0-4A71-BC19-EFB749C80293 (Deinocrotonidae fam. nov.), A09FB03D-BB80-4200-A109-03E787B9D36E (*Deinocroton* gen. nov.), and 1C208D0B-8C5C-44EC-8477-BDAE043704B3 (*D. draculi* sp. nov.).

**Data availability.** The data reported in this paper are detailed in the main text and in the Supplementary Information.

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

## Acknowledgements

Thanks are due to J. García and C. Paradela (MNCN) for technical help; A.R. Schmidt (Göttingen University) and M. Speranza (Mykolab, NGO, Nuremberg) for help analysing the data; H. Klompen (Ohio State University) for comments on some anatomical characters during an early stage of the research; J. Háva (Czech University of Life Sciences) for comments on hastisetae affinity; J. Zigras for providing the Burmese ixodid specimen and H. Chen for finding and providing the rest of the tick specimens; and O. Sanisidro (Kansas University) for making the reconstructions. This study is a contribution to the project CGL2014-52163 of the Spanish Ministry of Economy and Competitiveness. R.P.F. is funded by a Research Fellowship from the Oxford University Museum of Natural History.

## Author contributions

E.P., A.A., X.D., and R.P.-d.l.F. designed the project; E.P., S.R.A., and R.P.-d.l.F. performed the technical work; E.P. and R.P.-d.l.F. prepared the figures. All authors analysed the data and contributed to the discussion. E.P., A.A., D.A.G., and R.P.-d.l.F. wrote the manuscript.

## Additional information

**Competing interests:** The authors declare no competing financial interests.

**Change history:** The originally published version of this Article was updated shortly after publication to add the word 'Ticks' to the title, following its inadvertent removal during the production process. This has now been corrected in both the PDF and HTML versions of the Article.

