## [Peer Review File · Nature Communications]

Reviewers' comments:

Reviewer #1 (Remarks to the Author):

This is an excellent and potentially important contribution which describes a new family of tick from Burmese amber which is probably close to the enigmatic living family Nutallellidae. The authors make a strong case for this (and another) tick having fed on feathered dinosaurs in the mid Cretaceous and the manuscript as a whole is likely to be of considerable interest to the fields of parasitology, arthropod palaeobiology and for our knowledge of the Burmese amber forest fauna in general.

The text is competently written by acknowledged experts in the field, the figures are excellent and the associated video derived from the CT scan is of very high quality.

I believe this manuscript merits inclusion in an interdisciplinary journal such as 'Nature Communications' and I would recommend publication, subject to mostly minor revision as noted in the comments below.

GENERAL

The authors may not have seen the recently published paper:

Chitimia-Dobler et al. (2017). *Amblyomma birmittum* a new species of hard tick in Burmese amber. *Parasitology*. Available online.

which may need to be referenced too.

There are quite a lot of figures. I'm not sure what the limit is for this journal, but for example Fig. 2 while being a nice application of confocal microscopy, does not seem to add much to the overall interpretation.

Perhaps Fig. 8 would be better as a supplementary figure?

MINOR CORRECTIONS

line 22: add "gen et sp. nov." to indicate it is a new taxon ?

line 46: better "..entrapment near to the hosts's nest." ?

lines 67-69: So can festoon counts change through ontogeny? Adding a statement to this effect would support the notion that the two Poinar species may be synonyms, otherwise it reads a little bit strange to assign the fossil to *C. burmanicum* but then to state its most like *C. vetulum*....

line 76: "...allows visualisation of..."

line 126: Some data (e.g. absence of festoons) in the description is repeated from the, rather long, diagnosis. Please try to keep these separate.

line 277: "...do not directly..."

line 285: It might be worth noting comments (I think in one of Mans et al's papers?) that Poinar's extinct genera resemble Amblyomma; a genus which today uses a range of hosts, perhaps with a tendency towards reptiles.

line 344: "..became attached to their bodies."

line 360: I agree that the authors have good evidence that Deinocroton fed on feathered dinosaurs, but they do resemble Nutalliellids (which are now known to be generalists). I can see the advantage of popularizing the study by claiming to have a 'dinosaur parasite', but you might want to qualify this here by saying the new genus probably had feathered dinosaurs among its host spectrum; but may not have been an exclusive parasite of dinosaurs and/or early birds.

line 492: "in the tick..." [not "Tick"] ?

line 504-518: Titles of some papers here have words beginning with capitals. Please check Instructions to authors for the correct reference style.

line 730: Please check if Carios is now a synonym of Ornithodoros.

Fig. 10: I think it's misleading to characterize ALL Mesostigmata and 'carnivorous' as this conceals a wide diversity of feeding ecologies in this group from pollen-feeding to active predators to several parasitic lineages which (like ticks) are blood feeding ectoparasites too. I'd be tempted to leave out "Carnivorous" and "Liquid Feeders" from the graphic or to qualify these categories in some way.

line 732: The Baltic amber Hyalomma appears to be a misidentified rake-legged mite (Caeculidae) and should not be included in the tick fossil record. Please consult the recent (2017) Chitimia-Dobler paper for details.

line 841: "...on the one hand..."

line 969: "The ticks of..." ? [not Ticks]

line 1170: "...a new species..." [not Species]

line 1214: "...early Cretaceous..." [not cretaceous]

Reviewer #2 (Remarks to the Author):

Interesting paper describing new fossils from Burmese amber. The descriptions are very general, but this is often the case with fossils. The conclusions on host associations seem acceptable given the evidence, although I would certainly not consider that evidence overwhelming. Still, the conclusions generally fit within the generally accepted hypotheses on tick evolution. Perhaps a few more notes on the preliminary nature of the conclusions, but otherwise acceptable.

Most of my detail comments are relatively minor

The authors clearly are familiar with the structure of feathers, even as they appear to be a little less familiar with the anatomy of parasitiform mites.

For example patella may be correct in most Arachnida, but genu is the standard term used in mites. Metatarsus has not been used in mites for a long time. The basal secondary subdivision of tarsi II-IV is better described as basitarsus. I-171 "metatarsus and tarsus articulation in leg I not visible" reveals same the problem: there is never a distinct basitarsus on tarsus I in Parasitiformes (incl. ticks).

59 diagnostic characters in a tick larva do not always translate well to post-larval instars, so this statement seems somewhat optimistic. What diagnostic characters are shared? This is a somewhat loaded question as it was never very clear why this species was not included in Amblyomma (the "ventrolateral claw" on the palp genu is not convincing, the structure in question may be a set of thickened setae. De novo claws on the palp genu are quite unlikely).

94 legs "ruffled" What does that mean?

156 "swallow" spurs?

200 modification of palp femur is AN ADAPTATION to protect etc. I suggest a little more modesty. At the very least "might be an adaptation". I do not think sufficient data is available for a hypothesis of adaptation.

202 "... is present in Ixodidae...". In all Ixodidae or some Ixodidae?

223 I am not clear as to the logic of the argument made here. Please re-phrase.

287 Dermestidae not Desmestidae

Fig. 10 The phylogeny of Parasitiformes used is outdated. Current evidence suggest that Opilioacarida are the sistergroup to Holothyrida + Ixodida, with Mesostigmata as the first branch out. All Parasitiformes except Opilioacarida are fluid feeders, and both Opilioacarida and Holothyrida appear to be scavengers, not predators or parasites

I do applaud the approach by the authors of using dating for ticks based on fossil evidence, rather than models. At this point there is simply too little information to feel confident in model parameters. Their dating may be too conservative, but at least it seems based on real data.

Reviewer #3 (Remarks to the Author):

Dear Authors and Editor,
please find attached an annotated copy of the manuscript that provides suggestions for addressing minor issues within the work, and more specific advice for dealing with the issues outlined in this review.

This work does an exceptional job of outlining the first strong evidence for associations between ticks and their hosts in Burmese amber. The systematic and descriptive work on the ticks is comprehensive and commendable. There were only two significant issues with the work as it currently stands, and I recommend it for publication after minor/moderate revisions.

1) The description of the feather found associated with one of the ticks is detailed, but the interpretation appears to be a bit flawed. The authors have suggested that the feather has a more primitive morphology (Stage IIIa+b) than the figures support, with minor impact on their interpretation of which host groups it may represent.

- Based on the figures presented, it appears as though the feather in the new specimen is asymmetrical (both in terms of the barbs on either side of the rachis: Fig. 1a; and the barbules on either side of the barb: Figs. 1d, S1b-d). This, coupled with presence of a thick rachis basally would put the specimen in Stage IV or maybe even Stage V. This specimen looks very similar to a loosely vaned contour feather from a modern bird, and many contour feathers have simplified barbules as you approach the apex of each barb. (ca. line 262)

- The comparison to enantiornithine plumage works (these feathers are very similar to modern bird feathers based on recent studies from Burmese amber), but the coelurosaur tail plumage has a different micro- and macrostructure. The new specimen cannot be "identical" to both! In the coelurosaur feathers, the proximal barbules continue from the barbs down the side of the rachis, the barbules lack node spines, and the rachis is only slightly larger than the barb rami (in the new specimen it appears to thicken to nearly the width of an entire barb). Furthermore, the isolated barbule found in the new specimen (Fig. 1f) contains the transition from a blade-like base to a pennulum with hooklets -- this is diagnostic for a closed-vane pennaceous feather. It may have come from another feather, or a basal part of the contour feather being examined (ca. line 264)

- (around line 274): The pennulum with hooklets is also present in Stage V feathers, and has already been mentioned (but poorly illustrated) in work on enantiornithines flight feathers in Burmese amber (ref. 26). The feather or feathers represented in the new sample could belong to either Stage IV or V, and Stage V feathers with a similar morphology have already been described from the deposit.

- Constraining the feather to Stage IV or V does not impact the larger interpretation of these ticks being nidicolous in the nests of dinosaurs, but it suggests that you are dealing with an animal that falls within Pennaraptora, with Enantiornithes being one of the strongest candidates. Around line 817 in the SI, an excellent case is made for litter amber. This is neat because it suggests that you may be dealing with a nest on the ground, as has been suggested for Enantiornithes. It is worth mentioning this in the main text.

Some minor additional issues:

- for the rounded masses found associated with multiple joints on the legs, some justification should be presented for why these are not decay products escaping from arthroal membranes (line 338)

- Some parts of the description contain articles, while others do not, please adjust for consistency and brevity (line 120)

- around line 66: The current wording begs the question of why the new specimen is placed in one genus as opposed to the other. If the number of festoons is the only difference from *Compluriscutula*, but there are five differences, including a difference in festoon patterns from *Cornupalpatum* -- or this might just be a problem with word choice.

1. REVIEWER #1 (REMARKS TO THE AUTHOR)

This is an excellent and potentially important contribution which describes a new family of tick from Burmese amber which is probably close to the enigmatic living family Nutallessidae. The authors make a strong case for this (and another) tick having fed on feathered dinosaurs in the mid Cretaceous and the manuscript as a whole is likely to be of considerable interest to the fields of parasitology, arthropod palaeobiology and for our knowledge of the Burmese amber forest fauna in general.

The text is competently written by acknowledged experts in the field, the figures are excellent and the associated video derived from the CT scan is of very high quality.

I believe this manuscript merits inclusion in an interdisciplinary journal such as 'Nature Communications' and I would recommend publication, subject to mostly minor revision as noted in the comments below.

GENERAL

The authors may not have seen the recently published paper: Chitimia-Dobler et al. (2017). Amblyomma birmutum a new species of hard tick in Burmese amber. Parasitology. Available online. Which may need to be referenced too.

>Response. Done. We thank the reviewer for bringing this recently published work to our attention where a new species of ixodid tick from Burmese amber is described. This reference has been added to the main part of the manuscript (first paragraph of reworded C. burmanicum's remarks, see below), and the species has been added to Fig. 10's caption.

There are quite a lot of figures. I'm not sure what the limit is for this journal, but for example Fig. 2 while being a nice application of confocal microscopy, does not seem to add much to the overall interpretation. Perhaps Fig. 8 would be better as a supplementary figure?

>R. It is our understanding that the flexible formatting of Nature Communications allows for "up to 10 display items" in the main part of the paper depending on the word count (about 5,500 in our case, not counting references), and that the addition of 10 display items in the main part increases the interest and impact of the paper. Please note that Fig. 2, due to the application of confocal laser scanning microscopy and unlike other figures, shows critical morphological features of the ixodid nymph entangled with the feather, such as the lack of a genital pore in the specimen, which allowed us to determine that it is an immature (nymph). Likewise, Fig. 8 shows the rounded masses of amber in the distal leg segments of one deinocrotonid, which provide important taphonomic information to argue that the two deinocrotonid ticks preserved together were most likely caught by resin nearby the host's nest (lines 328 to 344). In any case, if the editorial board finds our figuration count excessive, we do not have any problem moving figs 2 and 8 to the supplementary information.

MINOR CORRECTIONS

line 22: add "gen et sp. nov." to indicate it is a new taxon ?

>R. Done

line 46: better "..entrapment near to the hosts's nest." ?

>R. We prefer to change the sentence to "... entrapment in close proximity to the host's nest."

lines 67-69: So can festoon counts change through ontogeny? Adding a statement to this effect would support the notion that the two Poinar species may be synonyms, otherwise it reads a little bit strange to assign the fossil to *C. burmanicum* but then to state its most like *C. vetulum*....

>R. Done. This wording issue was also pointed out by Reviewer #3. The number of festoons can (but do not necessarily) increase through ontogeny in hard ticks, according to Kleinjan and Lane (2008, "Larval keys to the genera of Ixodidae (Acari) and species of Ixodes (Latreille) ticks established in California." *The Pan-Pacific entomologist* 84.2 (2008): 121-142): "... Festoons (9 or 11 in larval Metastriata, 3 in Prostriata) sometimes are fused and appear fewer in number in larval ticks than in nymphs and adults, but are consistent within species." In any case, we have modified the first paragraph of the remarks for *C. burmanicum*, taking into account the suggestions from the second reviewer as well (see below).

line 76: "...allows visualisation of..."

>R. We think the the current version of the sentence is easier to read.

line 126: Some data (e.g. absence of festoons) in the description is repeated from the, rather long, diagnosis. Please try to keep these seperate.

>R. We acknowledge the concern of the reviewer for space, but as repeating the diagnostic characters throughout the description (and briefly expanding them therein) is the common practice in systematic works (as the diagnosis and the description fulfill different roles in systematics), and not following that convention would only be justified from our point of view if space was absolutely limited, we prefer to keep the description as it is, pending editorial approval. Additionally, please note that all the characters listed in the diagnosis are highly significant in distinguishing the new family from the modern tick families, as the Supp. Table 1 shows.

line 277: "...do not directly..."

>R. Done

line 285: It might be worth noting comments (I think in one of Mans et al's papers?) that Poinar's extinct genera resemble *Amblyomma*; a genus which today uses a range of hosts, perhaps with a tendency towards reptiles.

>R. Done, the resemblance of Poinar's genera with *Amblyomma* has been noted in the reworded first paragraph of *C. burmanicum*'s remarks (see below), and the Mans et al., 2012 reference where this was stated has been added to the main text (it was already in the supplementary information).

line 344: "...became attached to their bodies."

>R. Done

line 360: I agree that the authors have good evidence that *Deinocroton* fed on feathered dinosaurs, but they do resemble nutalliellids (which are now known to be generalists). I can see the advantage of popularizing the study by claiming to have a 'dinosaur parasite', but you might want to qualify this here by saying the new genus probably had

feathered dinosaurs among its host spectrum; but may not have been an exclusive parasite of dinosaurs and/or early birds.

>R. Done. Note that this matter was already tackled at the end of the concluding remarks when discussing why Deinocrotonidae went extinct, L. 373 "... deinocrotonids went extinct during the Cretaceous, possibly at the K–Pg extinction event, together with their feathered dinosaur hosts if these ticks were host specialists (Fig. 10). In any case, host specificity is not considered an important factor in the evolution of ticks, and many extant species are generalists, including *Nuttalliella namaqua*." To further address this, we have replaced "as" by "among" between "dinosaurs" and "their hosts" in L. 331, so that it reads "deinocrotonids most likely included feathered dinosaurs among their hosts".

line 492: "in the tick..." [not "Tick"] ?

>R. Note that the original publication writes tick capitalised, as indicated.

line 504-518: Titles of some papers here have words beginning with capitals. Please check Instructions to authors for the correct reference style.

>R. The paper titles here have been cited as they appear in their original publications, see the "how to submit" online document (<http://www.nature.com/ncomms/submit/how-to-submit>), "... the title should be written exactly as it appears in the work cited..."

line 730: Please check if *Carios* is now a synonym of *Ornithodoros*.

>R. Although not universally accepted and still requiring further assessment, the genus *Carios* was raised from the subgenus *Argas* (*Carios*) by Klompen & Oliver (1993) and contained 7 former genera and subgenera (see Burger et al., 2014). In any case, the valid combination to date for the species noted appears to be *Carios jerseyi* Klompen & Grimaldi, 2001, as also indicated by the fossil arachnid catalog of Dunlop et al., 2017 (supp. ref. 70).

Fig. 10: I think its misleading to characterise ALL Mesostigmata and 'carnivorous' as this conceals a wide diversity of feeding ecologies in this group from pollen-feeding to active predators to several parasitic lineages which (like ticks) are blood feeding ectoparasites too. I'd be tempted to leave out "Carnivorous" and "Liquid Feeders" from the graphic or to qualify These categories in some way.

>R. Done. The reviewer was correct in noting that the feeding categories that we were using did not encompass all the diversity of the groups. Changes in figure 10 have been made following a new reference now provided in Supplementary Information. Thus, in L. 936, the following has been added "Feeding ecology of Parasitiformes is after Walter & Proctor⁷¹." The latter reference has been added to the list of Supplementary References. Feedings habits of Mesostigmata are now represented as "D.F.H" (= "diverse feeding habits" to simplify their predatory, scavenging and parasitic habits), Opilioacarida appear as "scavengers, solid feeders", and Holothyrida are "scavengers, fluid feeders". Abbreviations for the latter have been used due to space restrictions in Fig. 10, and their meaning is provided in the figure caption. All the changes above are consistent with the notes provided by reviewers 1 and 2.

line 732: The Baltic amber *Hyalomma* appears to be a misidentified rake-legged mite (Caeculidae) and should not be included in the tick fossil record. Please consult the recent (2017) Chitimia-Dobler paper for details.

>R. Done. "*Hyalomma* sp." has been removed from the figure caption 10.

line 841: "...on the one hand..."

>R. Done.

line 969: "The ticks of..." ? [not Ticks]

>R. "Ticks" is capitalised in the original publication. See the "how to submit" online document (<http://www.nature.com/ncomms/submit/how-to-submit>), "... the title should be written exactly as it appears in the work cited...".

line 1170: "...a new species..." [not Species]

>R. "Species" is capitalised in the original publication, but also "New" thus "New" has been capitalised in the manuscript.

line 1214: "...early Cretaceous..." [not cretaceous]

>R. Note that "Cretaceous" is not capitalised in the original publication.

2. REVIEWER #2 (REMARKS TO THE AUTHOR)

Interesting paper describing new fossils from Burmese amber. The descriptions are very general, but this is often the case with fossils. The conclusions on host associations seem acceptable given the evidence, although I would certainly not consider that evidence overwhelming. Still, the conclusions generally fit within the generally accepted hypotheses on tick evolution. Perhaps a few more notes on the preliminary nature of the conclusions, but otherwise acceptable. Most of my detail comments are relatively minor.

The authors clearly are familiar with the structure of feathers, even as they appear to be a little less familiar with the anatomy of parasitiform mites. For example patella may be correct in most Arachnida, but genu is the standard term used in mites. Metatarsus has not been used in mites for a long time. The basal secondary subdivision of tarsi II-IV is better described as basitarsus. l-171 "metatarsus and tarsus articulation in leg I not visible" reveals same the problem: there is never a distinct basitarsus on tarsus I in Parasitiformes (incl. ticks).

>R. Done. We thank the reviewer for calling these two terminological inaccuracies to our attention. "Patella" or "patellar" has been changed to "genu" or "genua" in L. 162, 163, 651, 667, 834. "Metatarsus" has been replaced by "basitarsus" in L. 170. The sentence "metatarsus and tarsus articulation in leg I not visible" has been removed in L. 170.

59 diagnostic characters in a tick larva do not always translate well to post-larval instars, so this statement seems somewhat optimistic. What diagnostic characters are shared? This is a somewhat loaded question as it was never very clear why this species was not included in *Amblyomma* (the "ventrolateral claw" on the palp genu is not convincing, the structure in question may be a set of thickened setae. De novo claws on the palp genu are quite unlikely).

>R. Done. The statement indicated by the reviewer has been removed from the reworded text in the first paragraph of *Cornupalpatum burmanicum* "Remarks". Now the text includes the new species of Cretaceous *Amblyomma* recently described by Chitimia-Dobler et al., 2017, and explicitly states the diagnostic characters that led to the classification of our tick. We now indicate that the tick described in our manuscript resembles the ticks of the extant genus *Amblyomma*, and we emphasize the need for revision of the systematic status of the Cretaceous ixodids, which is beyond the scope of our contribution and does not impact its

palaeobiological implications. Overall, the former lines 57 to 69 now read: “The specimen AMNH Bu JZC-F18, preserved in Burmese amber, is a nymph based on its eight legs and absent genital pore (Figs. 1 and 2). The tick, ca. 0.9 mm long from the posterior margin to the apex of hypostome, has ventrolateral claws on palpomere III, lacks eyes, has all coxae with spurs and shows 11 festoons (Figs. 1 and 2, Supplementary Fig. 2a). Within the current diversity of Cretaceous hard ticks, none of them described as a nymph, these characters classify AMNH Bu JZC-F18 within *Cornupalpatum burmanicum*, described based on two larvae¹². The scutum, the teeth in the hypostome, the Haller’s organ, and the striate integument were not visible in the holotype of *C. burmanicum*, likely due to the specimen’s state of preservation. In addition, the new specimen does not fit some of the characters in the original description of the species, some of which could represent ontogenetic variation: the ventrolateral claws in the third palpal segment are less developed, the central festoon is as wide as the others (not narrower), and the second palpal segment is more elongated. In any case, we acknowledge that *C. burmanicum* and *Compluriscutula vetulum*, the other Cretaceous ixodid species based on a larval stage¹³, show a high degree of similarity with ticks of the extant genus *Amblyomma*¹⁴, and a Cretaceous species within that genus based on an adult was recently named¹⁵. A future revision of the described Cretaceous hard ticks reevaluating all the critical characters is necessary to elucidate their true relationships.”

94 legs “ruffled” What does that mean?

>R. Done. For clarity, we have added “ridges” in L. 162 (description) and placed “ruffles” between parentheses, so that it reads “Femur, patella and tibia bearing a sculptured surface of transverse ridges (ruffles), ...”

156 “swallow” spurs?

>R. Done. “swallow” corrected to “shallow”

200 modification of palp femur is AN ADAPTATION to protect etc. I suggest a little more modesty. At the very least “might be an adaptation”. I do not think sufficient data is available for a hypothesis of adaptation.

>R. Done: changed “is” by “appears to be”

202 “... is present in Ixodidae...”. In all Ixodidae or some Ixodidae?

>R. Done: added “all” before “ixodids”

223 I am not clear as to the logic of the argument made here. Please re-phrase.

>R. Done. This sentence was rephrased. It now reads (L. 223): “Other ixodids, such as *Boophilus*, have posterior grooves due to the presence of adanal shields; however, since *Deinocroton* lacks any structure resembling this shield, the posterior section of the genital groove in the new family appears to be unique among ticks.”

287 Dermestidae not Desmestidae

>R. Done

Fig. 10 The phylogeny of Parasitiformes used is outdated. Current evidence suggest that Opilioacarida are the sistergroup to Holothyrida + Ixodida, with Mesostigmata as the first branch out.

>R. Done. Figure 10 has been changed accordingly and L. 856 to 858 have been modified so that they now read “Current evidence suggests that Mesostigmata is sister to the rest of Parasitiformes, with Opilioacarida being sister to the clade (Holothyrida + Ixodida)^{19,20}.”

Moreover, in L. 933, "... based on Mans et al.⁸." has been replaced by "... based on Klompen²⁰ and Mans et al.¹⁹."

All Parasitiformes except Opilioacarida are fluid feeders, and both Opilioacarida and Holothyrida appear to be scavengers, not predators or parasites

>R. Done. See above (reviewer #1 comments on Fig. 10) for changes added on this matter.

I do applaud the approach by the authors of using dating for ticks based on fossil evidence, rather than models. At this point there is simply too little information to feel confident in model parameters. Their dating may be too conservative, but at least it seems based on real data.

3. REVIEWER #3 (REMARKS TO THE AUTHOR)

Dear Authors and Editor,

please find attached an annotated copy of the manuscript that provides suggestions for addressing minor issues within the work, and more specific advice for dealing with the issues outlined in this review.

>R. We are very grateful for the reviewer's careful edits along the manuscript and general advice. An account of the edits that have been followed throughout the annotated PDF and those that have not can be find below.

This work does an exceptional job of outlining the first strong evidence for associations between ticks and their hosts in Burmese amber. The systematic and descriptive work on the ticks is comprehensive and commendable. There were only two significant issues with the work as it currently stands, and I recommend it for publication after minor/moderate revisions.

The description of the feather found associated with one of the ticks is detailed, but the interpretation appears to be a bit flawed. The authors have suggested that the feather has a more primitive morphology (Stage IIIa+b) than the figures support, with minor impact on their interpretation of which host groups it may represent. Based on the figures presented, it appears as though the feather in the new specimen is asymmetrical (both in terms of the barbs on either side of the rachis: Fig. 1a; and the barbules on either side of the barb: Figs. 1d, S1b-d). This, coupled with presence of a thick rachis basally would put the specimen in Stage IV or maybe even Stage V. This specimen looks very similar to a loosely vaned contour feather from a modern bird, and many contour feathers have simplified barbules as you approach the apex of each barb. (ca. line 262)

The comparison to enantiornithine plumage works (these feathers are very similar to modern bird feathers based on recent studies from Burmese amber), but the coelurosaur tail plumage has a different micro- and macrostructure. The new specimen cannot be "identical" to both! In the coelurosaur feathers, the proximal barbules continue from the barbs down the side of the rachis, the barbules lack node spines, and the rachis is only slightly larger than the barb rami (in the new specimen it appears to thicken to nearly the width of an entire barb). Furthermore, the isolated barbule found in the new specimen (Fig. 1f) contains the transition from a blade-like base to a pennulum with hooklets -- this is diagnostic for a closed-vane pennaceous feather. It

may have come from another feather, or a basal part of the contour feather being examined (ca. line 264)

>R Done. We no longer consider the feather as belonging to Prum's stage IIIa+b but as stage V. The reviewer was correct in pointing out the asymmetry of the feather, which is now noted in remarks and allows to safely assign the feather to Prum's stage V. Although such asymmetry is rather apparent, we had not taken it into account in the manuscript before because we had not previously found "hooklets" on the barbs attached to the feather (only the isolated one shown in Fig. 1f), which are evolutionarily-developmentally prior to the presence of asymmetrical vanes according to Prum's models (stage V). On that regard, the discovery of further barbule pennula with hooklets on the feather during this revision (which allowed us to determine that the isolated barbule with hooklets shown in Fig. 1f detached from the own feather, as the reviewer suspected) is important. Images of these new barbules with hooklets are now provided in Supplementary Fig. 1e, f. Also, in the figure 10 caption, "Isolated barbule from the same amber piece" has been replaced by "Detached barbule". Moreover, following the reviewer's comments, we have replaced the concept of "pennaceous feather with open vanes" by "relatively loosely vaned pennaceous feather", compatible with the ability of the vanes to close in some areas of the feather due to the presence of hooklets. Lastly, the similarity of the feather reported by us to the feather published by Xing et al. (2016) from a coelurosaurian tail has been removed from the discussion after the reviewer's accurate remarks.

Accordingly, the second paragraph of *C. burmanicum*'s Remarks has been edited to account for the above changes and now reads: "Most significantly, the hard tick has one leg entangled in the barb of a pennaceous feather with a rather thick rachis basally (Fig. 1 and Supplementary Fig. 1). Its preserved section is 19.4 mm long and shows over 50 preserved barbs, most of them attached to the rachis but with their apices lost at the surface of the amber. Those barbs that happen to be complete are much shorter on one side of the preserved rachis section than those on the other side (ca. 11 mm vs. 19.5 mm long). Some barbs show damage, which likely occurred before having become embedded in the resin (Supplementary Fig. 1a). The fine preservation of the barbules allows us to distinguish their blade-like bases and their pennula, which display spined nodes and internodes. Most nodes in a distal position along the barbs are well defined and show short spines that are (sub)equally developed on both sides of the barbule pennulum (Fig. 1d; Supplementary Fig. 1c, d). Some poorly-defined nodes present in more proximal-medial areas of the barb, however, show relatively long spines on one side of the pennulum that form hooklets (=hamuli) (Supplementary Fig. 1e, f). In addition, two isolated barbs from a different feather are close to the semicomplete one (Supplementary Fig. 1b), and a detached pennulum showing hooklets on one of its sides, ca. 0.6 mm long, is also present in the amber piece (Fig. 1f). Pigments indicating colour patterns have not been observed."

Likewise, the first half of the first paragraph of the discussion has been edited as well to account for the changes noted above and it now reads: "The relatively loosely vaned pennaceous feather that the hard tick described herein is grasping (Figs. 1 and Supplementary Fig. 1) shows barbule pennula with hooklets in some areas. This would assign the feather to stage IV in Prum's evolutionary-developmental model of the feather, but the clear length asymmetry between the barbs on either side of the rachis classifies it within stage V²⁷. Even though stage IV and V feathers have for the most part been inferred in the fossil record, namely in compression fossils through the presence of well-developed closed vanes, some directly visible instances of these stages in Cretaceous amber feathers were previously reported (although not figured or poorly so) bearing barbules with hooklets like the ones presented here²⁸⁻²⁹. These structures have not been described from other Cretaceous feathers

found in Burmese³⁰⁻³¹, Canadian³², or Spanish ambers.” Note that the reference Nascimbene et al., 2014 (#28) has been added to the manuscript.

The text “The tick is entangled with the feather’s barb in virtually the same orientation, indicating that both contacted the resin together after separation from the feathered host. Entrapment within different resin flows of the feather and the tick is implausible because the resin is a viscous medium in which the entanglement of both entities cannot occur by slow contact due to drift into that medium.” has been moved out of the first paragraph of the discussion for the sake of cohesion, and it is now shown in its own paragraph together with information on the feather with the tick contacting the resin at ground level as indicated by the reviewer (see below). Please note as well that the Supplementary Figure 1 and its caption have been modified following the changes indicated above.

(around line 274): The pennulum with hooklets is also present in Stage V feathers, and has already been mentioned (but poorly illustrated) in work on enantiornithines flight feathers in Burmese amber (ref. 26). The feather or feathers represented in the new sample could belong to either Stage IV or V, and Stage V feathers with a similar morphology have already been described from the deposit.

>R. We acknowledge what the reviewer points out here, although, as the reviewer indicated, the structures in question (hooklets) are not discernible from the images provided in Xing et al., 2016, Fig. 2e (ref. 26). The changes done to this part of the manuscript are provided in the point above.

Constraining the feather to Stage IV or V does not impact the larger interpretation of these ticks being nidicolous in the nests of dinosaurs, but it suggests that you are dealing with an animal that falls within Pennaraptora, with Enantiornithes being one of the strongest candidates. Around line 817 in the SI, an excellent case is made for litter amber. This is neat because it suggests that you may be dealing with a nest on the ground, as has been suggested for Enantiornithes. It is worth mentioning this in the main text.

>R. Please note that in the paper we only present evidence of nidicolity (i.e., living in the host’s nest or in a harborage nearby) for deinocrotonids, not for the ixodid attached to the feather (in fact, all extant ixodids but those within the genus *Ixodes* are known to be free living forms, so not nidicolous). Data for suggesting that the ixodid’s feathered host had its nest on the ground based on the record that we present is insufficient. In any case, the observation of the amber piece with the feather and the ixodid attached to it having originated at the ground level was previously only provided in Supplementary Information, and it now appears in the main text, also considering the option that the feather with the tick attached and/or the resin could have fallen from above, reading “Such contact most likely occurred at the ground level as indicated by the overall fossil assemblage preserved in the amber piece (see Supplementary Information), although both the feather carrying the tick and/or the resin that encased them could have fallen from above.” (this text is in the new second paragraph of the discussion, see above).

Some minor additional issues:

for the rounded masses found associated with multiple joints on the legs, some justification should be presented for why these are not decay products escaping from arthroal membranes (line 338)

>R. Instances of internal fluids escaping from leg joints without traumatism are lacking in the fossil record of amber, to our knowledge. There are some reported instances of fluid escaping in haemolymph droplets from autospasised legs (detached at a predetermined locus of weakness when restrained by a non-self-induced source) in two spiders (Penney, D. 2005. Fossil blood droplets in Miocene Dominican amber yield clues to speed and direction of resin secretion. *Palaeontology*, **48** (5): 925-927.). In these cases, however, the haemolymph droplets have a very different appearance than in the tick that we report, having an even surface (vs. rugose) and lacking internal bubbles (vs. showing abundant bubbles in the resin drops from the tick).

Abundant cases of the bee *Proplebeia dominicana* (Hymenoptera: Apidae) in Dominican amber are known in which they have “resin balls” preserved attached to specialised hindleg structures (corbiculae), resulting from the bees gathering fresh resin for their nests as known from related extant bees. The morphology of these resin balls is identical to that from the resin balls attached to the tick legs (see Poinar, 1992, Fossil evidence of resin utilization by insects. *Biotropica*, *24*(3), 466-468: Fig. 1). Note how the resin balls of the bees preserve multiple bubbles inside, in a similar fashion that is shown for the specimens that we present (as shown in Fig. 8c). Even though in one case the resin balls were collected and in the second case were an accidental consequence, in both cases the manner in which the resin became attached to the legs was very similar, due to contact of the legs with resin during the arthropod's movement.

Furthermore, although internal fluids escaping from the mouth and anus of insects have been documented in amber inclusions, in these cases the fresh resin has usually diluted these fluid internal emissions. Thus their morphology is significantly different than that seen in the “amber balls” of the tick, in which the edges are very well-defined, showing that the rounded resin masses entered the fresh resin already polymerised (hardened).

Additionally, if there was internal fluid escaping from the legs of the tick, one would expect the pressure of the fluid to be released only in a single or few points along the tick's legs or body, whereas in the specimen that we describe there are at least seven hardened resin droplets corresponding to the leg areas that initially made contact with the fresh resin.

Following the discussion above, we have added a brief clarification on this matter, adding in L. 333: “, abundantly reported from *Proplebeia* bees in Dominican amber⁴¹,” between “These amber drops” and “are distinct from...”. We believe that this reference is enough for the reader to assess the evident similar morphology between the amber drops described from our tick and those abundantly reported from the Dominican amber bees and, therefore, to rule out less plausible alternatives.

Some parts of the description contain articles, while others do not, please adjust for consistency and brevity (line 120)

>R. Done (see below)

8. around line 66: The current wording begs the question of why the new specimen is placed in one genus as opposed to the other. If the number of festoons is the only difference from *Compluriscutula*, but there are five differences, including a difference in festoon patterns from *Cornupalpatum* -- or this might just be a problem with word choice.

>R. Done. This wording problem has been solved (see above)

Edits suggested by Reviewer #3 (in the pdf) applied:

L. 42. Removed “ i.e.,” between “of its host,” and “a hard tick...” and inserted a “—“ (m-dash) in its place

- L. 43. Added “found” between “also” and “in Burmese amber,”
- L. 62. Replaced “can be due to bad preservation” to “could be due to poor preservation”
- L. 65. Removed “bad” between “related to” and “preservation”
- L. 66. Added “variability” between “preservation” and “or to the larval”
- L.66. Wording corrected: replaced “is more similar” by “shows a high degree of similarity”
- L. 92. Added “s” after “Spiracle”
- L. 101. Replaced “itself” by “which is”
- L. 113. Removed several “the” along the description for consistency and brevity. Please let us know if a more complete voice for the description of the new taxon would be advisable for the style of the journal. However, we have not removed “the” in the following cases: 1) In the expression “the rest of the body”, which we have condensed as “the rest of body) (not as just “rest of body”), 2) in superlatives (e.g. “the longest”, “the anteriormost part”), 3) before “paratype male”, 4) in comparative expressions, e.g, “two times the length”, 5) in the expression “without forming sockets for the articulation).
- L. 131. Replaced “, i.e.,” by “(“ . Closed the parentheses at the end of the sentence (L. 132)
- L. 139. Replaced “,” between “5g” and “quadrangular” by “that is”
- L. 141. Replaced “to” between “in” and “two sections” by “into”
- L. 152. Replaced “from anterior 2/5 of body” by “within anterior two-fifths of total body length”
- L. 153. Added a “,” after “*Nuttalliella*”
- L. 155. Replaced “Coxae slightly separated between them except to the” by “Slight separation between coxae, except”
- L. 157. Replaced “-“ (n dash) by “—“ (m dash).
- L. 164. Added “positioned” between “I and II” and “very high”
- L. 183. Replaced “closely relate” by “suggest a close relationship between.” Replaced “to” between “*Deinocrotonidae*” and “*Nuttalliellidae*” by “and”
- L. 203. Removed “1)” and “2)”, added a “,” replaced “and” by a “,” and changed “takes” to “taking” so that the sentence reads “... although palpomere III is also expanded, taking part in the protection of the gnathosoma, and both palpomeres are straight...”
- L. 213. Replaced “,i.e.,” by “(“, “lacks” by “lacking”, added a “,” after “*Ixodes*”. Note that instead of closing the parenthesis at the end of the sentence as suggested by reviewer, we suggest to do so after “transverse slit”
- L. 218. Replaced “,” by “(“ between “*Ixodidae*” and “i.e. bigger”, added a “,” after “i.e” and closed the parenthesis at the end of the sentence
- L. 220. Added “” enclosing “posterior genital groove sections”
- L. 223. Added two “,”, one between “Other ixodids” and “such as”, and “*Boophilus*” and “have posterior”
- L. 224. Replaced “, but as” by “; however,”. Replaced “the new family” by “*Deinocroton*”. Replaced “the latter” by “this shield”
- L. 235. Replaced “approximately” by “in”
- L. 243. Added “.” after “ca”. Added a “,” after “width”
- L. 246. Replaced “namely” by “particularly in”
- L. 306. Removed “with” between “contacted” and “the resin”
- L. 311. Removed “for” between “seek” and “hosts”
- L. 313. Replaced “the” by “an”
- L. 314, 315, 316. Replaced “,” after “nidicolyl”, “vertebrates”, and “wounds”, respectively, by a “;”
- L. 317. Added “s” after “host”

- L. 323. Replaced “, as ticks” by “—“, and added “ticks” between “occur,” and “voluntarily” (note. Reviewer suggested to use a “.” instead of the m-dash, but we think that the latter is better to maintain the relationship between the two sentences)
- L. 326. Replaced “discards” by “casts further doubt on the idea” (note: different version from that suggested by the reviewer)
- L. 335. Replaced “took” by “made”
- L. 341. Replaced “discard” by “undermines the idea”
- L. 348. Added a “,” after “its host”
- L. 350. Replaced “i.e., they fed” by “feeding”, and added “in” before “minutes”. The “had” before “multiple” also had to be replaced by “having” because of the former change
- L. 363. Added “st” after “late” so that it reads “latest”
- L. 368. Replaced “occurred” by “extend”, and added “and” after “likely earlier”
- L. 369. Added “that are” before “considered”
- L. 374. Replaced “on” by “of” after “evidence”
- L. 375. Added “,” after “ambers”
- L. 381. Replaced “and” by “where” and removed “therein”
- L. 406. Added “to have” between “appeared” and “slow”
- L. 408. Added “with” before “speed”, “and a” before “frame” and “of” before “4”
- L. 411. Replaced “SCAN” by “scan”
- L. 418. Added “by” between “prepared” and “modifying”
- L. 420. Replaced “performed” by “created”
- L. 428. Replaced “acceptance” to “acceptance”
- L. 604. Added “pennulum” after “isolated barbule”. “Pennulum” also added “after isolated barbule” in L. 78 for consistency
- Supp. Table 1. Char. Integument Setation: Replaced “to” by “with” between “associated” and “the pits”
- Supp. Table 1. Char. Eyes: Removed two “the”, one between “on” and “sides” and the other one between “of” and “scutum”
- L. 848. Replaced “, while” by “; meanwhile”
- L. 905. Replaced “allows to consider” by “suggests”
- L. 906. Added “pycnofibres” between “these” and “are not” (note the Oxford English spelling vs “pycnofibers”)
- L. 915. Added “,” between “debris” and “or in mammal”
- L. 946. Replaced “posteriorly” by “subsequently”

Edits suggested by Reviewer #3 (in the pdf) NOT applied:

- L. 21. We have not replaced “the ancestral new family” by “the newly described ancestral family” because “newly” can be understood as synonymous with “recently”, and therefore it would be left unclear whether the new family that we are describing has been already described elsewhere.
- L. 32. “Cainozoic” is the correct Oxford English spelling for Cenozoic. Also in lines 369 and 927.
- L. 44. The reviewer suggested to change “can” by “may” in the sentence “tick specimens of a new family, also in Burmese amber, can be indirectly related to feathered dinosaur hosts due to the presence of specialised setae from dermestid beetle larvae (hastisetae) attached to them and further evidence of taphonomic nature, both indicating resin entrapment in close proximity to the host's nest.” However, we think that the uncertainty in this statement is sufficiently stressed by the word “indirectly”, and at the same time the (indirect) evidence for the inference of the tick belonging to the new family is strong enough to preclude using “may”.

L. 64 – It is our understanding that the adverb “well” is only hyphenated together with the adjective that complements when “well + adjective (e.g., developed, defined)” appears before the noun to which they complement. E.g. “a well-developed organ”, vs “a organ is well developed”. Also noted by reviewer in lines 127, 140, 144, 174.

L. 203 and 204. We did not replace the “,” by “;”

L. 282. The reviewer recommended to add the sentence “, but basal avialans cannot be excluded based on the evidence currently available.” after “... the combination of feathers belonging to the stages IIIa+b and IV indicates that the dinosaur host of the hard tick described herein falls within the clade Pennaraptora according to current evidence from the fossil record of feathered dinosaurs (see Supplementary Information). Crown-group birds are excluded as possible hosts as their inferred age is significantly younger than Burmese amber, i.e., about 73 Ma based on targeted next-generation DNA sequencing.” arguing that the addition made the discussion match the abstract. However, by indicating that the dinosaurs hosting the tick that we describe fall within the clade Pennaraptora we are also taking into account that the hosts could have been basal avialans (as Pennaraptora is a clade basal to Avialae). Note that the possibility of our hosts being basal avialans is also included in the abstract “Here we report direct and indirect evidence in 99 million-year-old Cretaceous amber showing that hard ticks and ticks of the ancestral new family †Deinocrotonidae fed on blood from feathered dinosaurs, non-avialan or avialan excluding crown-group birds”, and also in L. 362 (concluding paragraph of the discussion): “Direct evidence herein proves that hard ticks fed on blood from feathered theropods (non-avialan or avialan) during the late Early Cretaceous, [...]. Deinocrotonids most likely were ectoparasitic on feathered dinosaurs (Fig. 9) as well based on...”, and also showcased in Fig. 10. Therefore, we think that the addition noted by the reviewer is unnecessary.

L. 330. The reviewer suggested to replace “allows to infer” by “suggests” in the sentence “The presence of dermestid hastisetae in the two *Deinocroton* preserved together and the inferred nidicolous ecology of these ticks, when considering the scarce record of hairs³⁸ vs. that of feathers^{26–28} in Cretaceous amber (particularly in Burmese amber), allows us to infer that deinocrotonids most likely had feathered dinosaurs as their hosts”. However, the uncertainty of our inference is already expressed in the expression “most likely” above, so that adding “suggest” would provide too much uncertainty for an inference that is, as the paper discusses, well backed up by data of diverse nature. Note that “us” has been added between “allows” and “to infer”.

L. 366. A “,” has not been added after “as well” because we think it is unnecessary

L. 372. Added “event,” between “extinction” and “together”

L. 443. “Flea” and “Pulicid” capitalised in original publication

L. 509. Idem: most words are capitalised in original publication

L. 517. Idem: most words are capitalised in original publication

L. 729. “attached to” has been replaced by “entangled in” instead of “associated with”, as the reviewer was suggesting, which is the expression used in the abstract and one that better reflects the interaction between the ixodid tick and the feather in the amber piece.

L. 876. Instead of removing the “,” between “detail” and “as chert deposits”, we have added “such” before “as chert deposits”

L. 969. “Ticks” capitalised in original publication

L. 973. Idem: most words are capitalised in original publication

L. 979. “*Ticks tick*” is not duplicate. The journal that is abbreviated is “*Ticks and tick-borne diseases*”

L. 995. “Estimation” capitalised in original publication

L. 1026. “Period” capitalised in original publication

L. 1064. Idem: most words are capitalised in original publication

L. 1068. Name of webpage does not have “of the”. See www.dermestidae.wz.cz/main.html.

L. 1070. “Species” capitalised in original publication

4. ADDITIONAL EDITS

L. 14. Please note the change of affiliation for the last author: “Museum of Comparative Zoology, Harvard University, Cambridge, 02138 MA, USA.” has been replaced by “Oxford University Museum of Natural History, Parks Road, Oxford, OX1 3PW, UK.”

L. 39. Added “the” between “appears to be” and “sister group”

L. 45. Replaced “them and” by “the ticks, along with”

L. 58. Replaced “absence of” by “absent”

L. 207. Replaced “carries out” by “provides”

L. 212. Replaced “been” by “remained”

L. 218. Replaced “to” by “from”

L. 222. Reworded “similar to the position and extension of the genital grooves” so that it reads “similar in position and extension to the genital grooves”

L. 232. Replaced “that” by “than”

L. 236. Replaced “to” by “from”

L. 241. Added parentheses between “attributed to engorgement”

L. 244–252. Replaced “,” by “;” in elements of the enumeration

L. 249. Replaced “between” by “from”

L. 281. Replaced “as” by “because”

L. 285. Replaced “had further” by “also parasitised other”

L. 286. Removed bold from “spear”

L. 294. Replaced “scape” by “escape”

L. 298. Replaced “affiliates them to” by “shows that they are most likely affiliated with”

L. 300. Added “a” before “commensalistic relationship”

L. 302. Replaced “sustain” by “sustains”

L. 319. Replaced “its” by “the” between “had been in” and “host’s nest”

L. 326. Replaced “its” by “their” before “host”

L. 332. Replaced “the” by “its” between “some of” and “leg apices”; replaced “from” by “on”

L. 333. Replaced “by” by “due to” between “matrix” and “darker colour”

L. 334. Added “likely” between “the tick” and “first”

L. 336. Replaced “in” by “with” between “coated” and “resin”

L. 338. Added “its” between “both” and “first legs”

L. 347. Replaced “in addition” by “also”

L. 350. Replaced “as” by “like that observed” between “feeding was” and “in nuttalliellids”

L. 352. Replaced “as” by “since” between “(Supplementary Table 3),” and “their adult”

L. 368. Replaced “then” by “at least that time” between “back to” and “for ticks”

L. 386. Replaced “its blood meal had finished” by “it had completed its blood meal”

L. 584. Added “(Mykolab, NGO, Nuremberg)” after “Speranza”

L. 590. Added “RPF is funded by a Research Fellowship from the Oxford University Museum of Natural History.”

L. 855. Removed “in” between “There is consensus” and “that”

References have been renumbered after the addition of three references in the main text, and one reference in the Supplementary Information.

Thank you.

REVIEWERS' COMMENTS:

Reviewer #3 (Remarks to the Author):

The authors have satisfactorily addressed all of the issues raised in my initial review. It also appears as though all of the issues raised by the other two reviewers have been addressed as well. I now recommend the manuscript for publication.